# Four distinct network patterns of supramolecular/polymer composite hydrogels controlled by formation kinetics and interfiber interactions

Keisuke Nakamura [1], Ryou Kubota [1] ✉, Takuma Aoyama[2,3], Kenji Urayama[3] & Itaru Hamachi [1,4] ✉

Synthetic composite hydrogels comprising supramolecular fibers and covalent polymers have attracted considerable attention because their properties are similar to biological connective tissues. However, an in-depth analysis of the network structures has not been performed. In this study, we discovered the composite network can be categorized into four distinct patterns regarding morphology and colocalization of the components using in situ, real-time confocal imaging. Time-lapse imaging of the network formation process reveals that the patterns are governed by two factors, the order of the network formation and the interactions between the two different fibers. Additionally, the imaging studies revealed a unique composite hydrogel undergoing dynamic network remodeling on the scale of a hundred micrometers to more than one millimeter. Such dynamic properties allow for fracture-induced artificial patterning of a network three dimensionally. This study introduces a valuable guideline to the design of hierarchical composite soft materials.

Living tissues are mechanically supported by the extracellular matrix (ECM) to achieve vital functions[1,2]. ECM comprises a multicomponent network of various biopolymers ranging from fibrillar collagen proteins to proteoglycans covalently modified with glycosaminoglycan chains. Each network plays a different role in the tissue's mechanical and structural properties. In cartilage, for example, the collagen fibers form a rigid network to resist tension, while proteoglycans with polyanionic chondroitin sulfate chains can accumulate water to resist compression[3,4]. Moreover, recent biophysical studies suggest that the hierarchical ECM network formed through the interplay between individual components allows for new properties such as nonlinear mechanical responses and dynamic network remodeling[5,6]. Such

structures and mechanics of these composite networks provide a valuable baseline for the design of new synthetic soft materials with life-like features.

Synthetic polymer hydrogels with two types of polymer networks have been intensely developed in recent years[7,8]. Gong et al. pioneered chemically crosslinked double network (DN) hydrogels containing two polymer networks with distinct physicochemical characteristics, a rigid electrolyte network and ductile neutral network[9–12]. Because of energy dissipation by the brittle network, the DN hydrogels can have high strength and toughness that are nearly comparable to load-bearing tissues such as cartilage. Recently, the concept of the DN hydrogel has been extended by replacing the brittle network with

[1]Department of Synthetic Chemistry and Biological Chemistry, Graduate School of Engineering, Kyoto University, Kyoto, Japan. [2]Department of Macromolecular Science and Engineering, Kyoto Institute of Technology, Kyoto, Japan. [3]Department of Material Chemistry, Graduate School of Engineering, Kyoto University, Kyoto, Japan. [4]JST-ERATO, Hamachi Innovative Molecular Technology for Neuroscience, Kyoto, Japan. ✉e-mail: rkubota@sbchem.kyoto-u.ac.jp; ihamachi@sbchem.kyoto-u.ac.jp

reversible physical networks, such as a self-assembled lamellar bilayer and physically crosslinked polymers, to achieve self-healing and fatigue resistance[13–15]. Other attractive multiple networked soft materials are composite hydrogels comprising covalent polymers and supramolecular nanofibers[16]. Supramolecular nanofibers are formed by the self-assembly of synthetic low molecular weight (LMW) gelators via non-covalent interactions[17,18]. The programmable design of LMW gelators allows for the flexible implementation of stimuli responses into supramolecular nanofibers[19–29]. Supramolecular/polymer composite hydrogels are promising matrices for the controlled release of biopharmaceuticals because of the distinct roles of the two networks such as stimulus responses and protein entrapment by the supramolecular fibers and mechanical stiffness by the covalent polymers[30–41]. To clarify and control the structures/properties of such composite hydrogels, several researchers have attempted to visualize the network structures using microscopic imaging techniques such as scanning electron microscopy (SEM) and confocal laser scanning microscopy (CLSM)[42]. Smith and coworkers reported that agarose and a sugar-derived LMW gelator self-assemble without interference from each other to form orthogonal nanofibers, as confirmed by SEM[34]. We succeeded with in situ CLSM imaging of the orthogonal network of agarose and peptide-type supramolecular fibers[40]. Alternatively, van Esch et al. demonstrated the colocalized network of calcium alginate and supramolecular nanofibers using CLSM[37]. However, comprehensive studies on the network structures of supramolecular/polymer composite hydrogels have not been performed. Therefore, the type of network patterns in the composite and, more importantly, the factors controlling the network patterns, are not clear.

Here, we investigated the network structures of supramolecular/polymer composite hydrogels comprising various types of LMW gelators. Super-resolution CLSM imaging identified four distinct network patterns that were classified on the basis of the morphology and colocalization of the components (Fig. 1a). Real-time in situ CLSM imaging of the formation processes revealed that the network patterns are mainly controlled by two factors: the order of network formation and interactions between the supramolecular gelators/fibers and the polymer. Furthermore, we identified a unique composite hydrogel that undergoes a time-dependent network change from a homogeneously distributed pattern to a phase-separation-like pattern at sub-millimeter scale. Such dynamic property induced the fracture-triggered remodeling of the composite network such that the artificial pattern was fabricated inside the composite hydrogel in two and three dimensions.

## Results
### Exploration of the network patterns in supramolecular/polymer composite hydrogels
We previously developed a supramolecular/polymer composite hydrogel comprising APmoc-F(CF$_3$)F, a diphenylalanine-based LMW gelator bearing an acetoxyphenyl group at the N-terminus, and agarose, a physically crosslinked polysaccharide hydrogel (Supplementary Fig. 1)[40]. In situ CLSM imaging revealed an orthogonally segregated structure of the APmoc-F(CF$_3$)F nanofibers and the agarose network whose morphologies were identical to those of single-component hydrogels. In this study, we investigated the network structures of composite hydrogels comprising agarose and structurally varied LMW gelators. Three peptide-type LMW gelators with a di- or triphenylalanine sequence and a distinct N-terminal moiety [boronophenyl (BPmoc[22]), nitrophenyl (NPmoc[26]), and benzaldehyde (Ald) derivatives[43]] and three lipid-type gelators with the different head groups [phosphate (Phos)[43], GalNAc[44], and Lys[45]] were picked up from our small library of LMW gelators, as shown in Fig. 1b and Supplementary Fig. 1. The peptide-type gelators self-assemble into β-sheet like nanofibrous structures via hydrogen bonding and π–π interactions between the phenylalanine peptides[46]. The lipid-type gelators are amphiphiles comprising a hydrophilic head and hydrophobic

cycloalkane tail group that form one-dimensional nanofibers mainly via hydrophobic interactions in an aqueous solution[47]. According to our previous report[48], the head groups locate at the surface of supramolecular fibers and are exposed to the water phase. We thus examined the effects of the head groups rather than the hydrophobic tail groups, which are buried inside the nanofibers, on the composite network structures through the interaction with the agarose network. We also employed DBS-COOH, a sugar-derived LMW gelator that was originally developed by Smith's group[33,34]. Agarose was used as a representative polymer-based hydrogel. Agarose forms a three-dimensional network of aggregated double helix structures via hydrogen bonding as physical crosslinks[49]. Both the supramolecular nanofibers and agarose network were fluorescently stained for CLSM imaging, as previously reported (Fig. 1c)[40]. As a fluorescent probe for the peptide-type hydrogelators and DBS-COOH, TMR-Gua was used to selectively interact with the carboxylate moiety of the nanofibers using its guanidium moiety. The lipid-type nanofibers were fluorescently stained by Alx546-cycC$_6$, which has a hydrophobic self-assembling moiety similar to the lipid-type gelators[50]. For the polymer network, agarose was covalently modified with Alexa Fluor 488 (Alx488-Agarose)[51].

### Four distinct network patterns of supramolecular/polymer composite hydrogels visualized by CLSM imaging
We initially examined the network structures of single-component hydrogels. A suspension of the LMW gelator or Alx488-Agarose powder in a buffer solution (100 mM MES, pH 7.0) containing the corresponding fluorescent probes was heated until dissolved, followed by cooling to room temperature (rt) to form a hydrogel (termed a heat–cool protocol). CLSM imaging illustrated that both peptide- and lipid-type hydrogelators self-assembled into well-developed networks comprising entangled nanofibers with a diameter of ca. 100–200 nm (Fig. 2a and Supplementary Fig. 2). Alx488-Agarose showed a sea–island network, corresponding to an aggregated double helix structure (Fig. 2a and Supplementary Fig. 2). To quantitatively analyze an agarose network structure, we estimated the average sizes of the island domains and void spaces by particle analyses ($0.27 \pm 0.02\ \mu m^2$ and $0.51 \pm 0.03\ \mu m^2$, respectively) (See Methods for detail; Supplementary Fig. 3). The homogeneity of the agarose network was also quantified by histogram analyses to evaluate the variance of the fluorescence intensity distribution (see Methods for detail; Supplementary Fig. 4).

We next investigated the network structures of the supramolecular/agarose composite hydrogels. The composite hydrogels were prepared from a mixture of the LMW gelator, a corresponding fluorescent probe, and Alx488-Agarose using the heat–cool protocol. In the case of BPmoc-F$_3$/Alx488-Agarose, CLSM illustrated the fibrous morphology of BPmoc-F$_3$ and the sea–island network of agarose, whose morphologies were nearly similar to those of the single-component hydrogels (Fig. 2a, b, Supplementary Figs. 3 and 4). The overlay image showed that the two networks were not correlated with each other; the nanofibers were mainly located in the darker regions (void spaces) of the agarose network. This observation was supported by the line plot analysis, which showed that the fluorescence peaks did not overlap with each other (Fig. 2b). The low average Pearson's correlation coefficient[52] value ($0.08 \pm 0.10$) suggested negligible correlation between the localization of BPmoc-F$_3$ and Alx488-Agarose. Therefore, the composite hydrogel of BPmoc-F$_3$/agarose is an orthogonal network, similar to the previously reported APmoc-F(CF$_3$)F/agarose hydrogel (Fig. 2c)[40].

In a GalNAc-cycC$_6$/Alx488-Agarose composite, a network pattern distinct from the orthogonal network was observed, where the morphology of the agarose was substantially altered from that of its single component. Specifically, agarose formed a fibrous network unlike the original sea–island pattern (Fig. 2d), whereas GalNAc-cycC$_6$ showed a

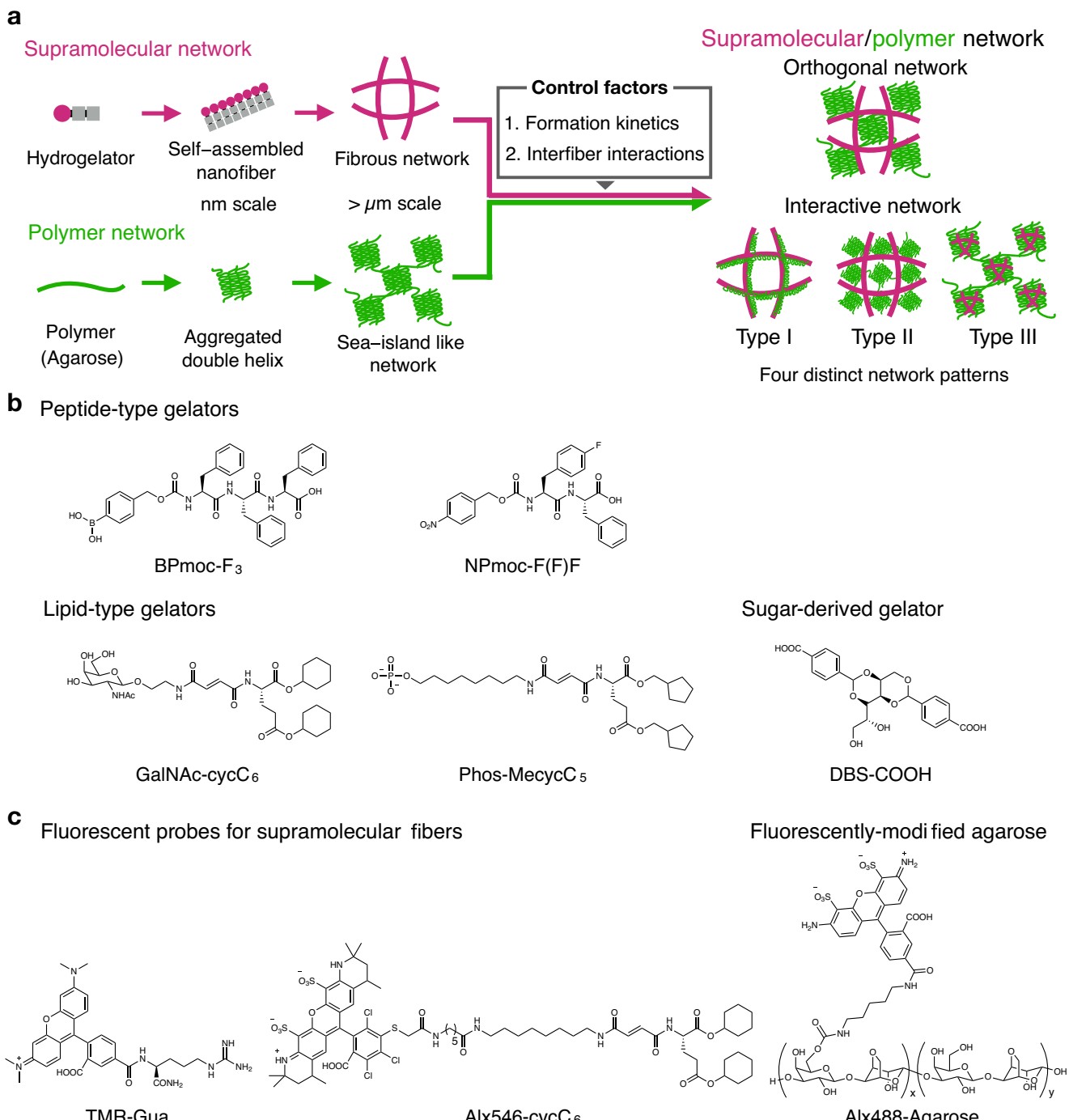

**Fig. 1 | Four distinct network patterns of supramolecular/polymer composite hydrogels. a** Schematic illustration of four distinct network patterns in composite hydrogels. **b** Chemical structures of representative peptide-type, lipid-type, and sugar-derived hydrogelators. Chemical structures of other hydrogelators are shown in Supplementary Fig. 1. **c** Chemical structures of fluorescent probes for supramolecular nanofibers and fluorescently-modified agarose.

fibrous morphology similar to that of the single component (Fig. 2d). Interestingly, the fiber-like agarose network merged well with the GalNAc-cycC₆ nanofibers, and the line plot analysis showed well-overlapped peaks of GalNAc-cycC₆ and agarose. The average nearest peak distance between the supramolecular and agarose network was estimated to be $120 \pm 110$ nm, which was statistically smaller than that of the orthogonal BPmoc-F₃/Alx488-Agarose ($200 \pm 200$ nm) (Supplementary Fig. 5, 6, 9 and Supplementary Table 1). The average Pearson's correlation coefficient ($0.38 \pm 0.12$) was greater than that of the orthogonal network (BPmoc-F₃/agarose: $0.08 \pm 0.10$). The interactions between the GalNAc-cycC₆ nanofibers and agarose may change

the network morphology of agarose. This network is referred to as interactive network type I (Fig. 2e).

The composite hydrogel of Phos-MecycC₅ and Alx488-Agarose exhibited another type of interactive network, where the void size of the agarose sea–island network decreased to $0.32 \pm 0.03\ \mu m^2$ and formed a more uniformly distributed structure than that of the single component (Fig. 2f, and Supplementary Fig. 3). The standard deviation ($s$) of the histogram of the Alx488 channel in the Phos-MecycC₅/agarose [$s = (1.264 \pm 0.012) \times 10^4$] was less than that of the single-component agarose [$s = (1.373 \pm 0.010) \times 10^4$] or the orthogonally entangled BPmoc-F₃/agarose [$s = (1.365 \pm 0.007) \times 10^4$], suggesting the

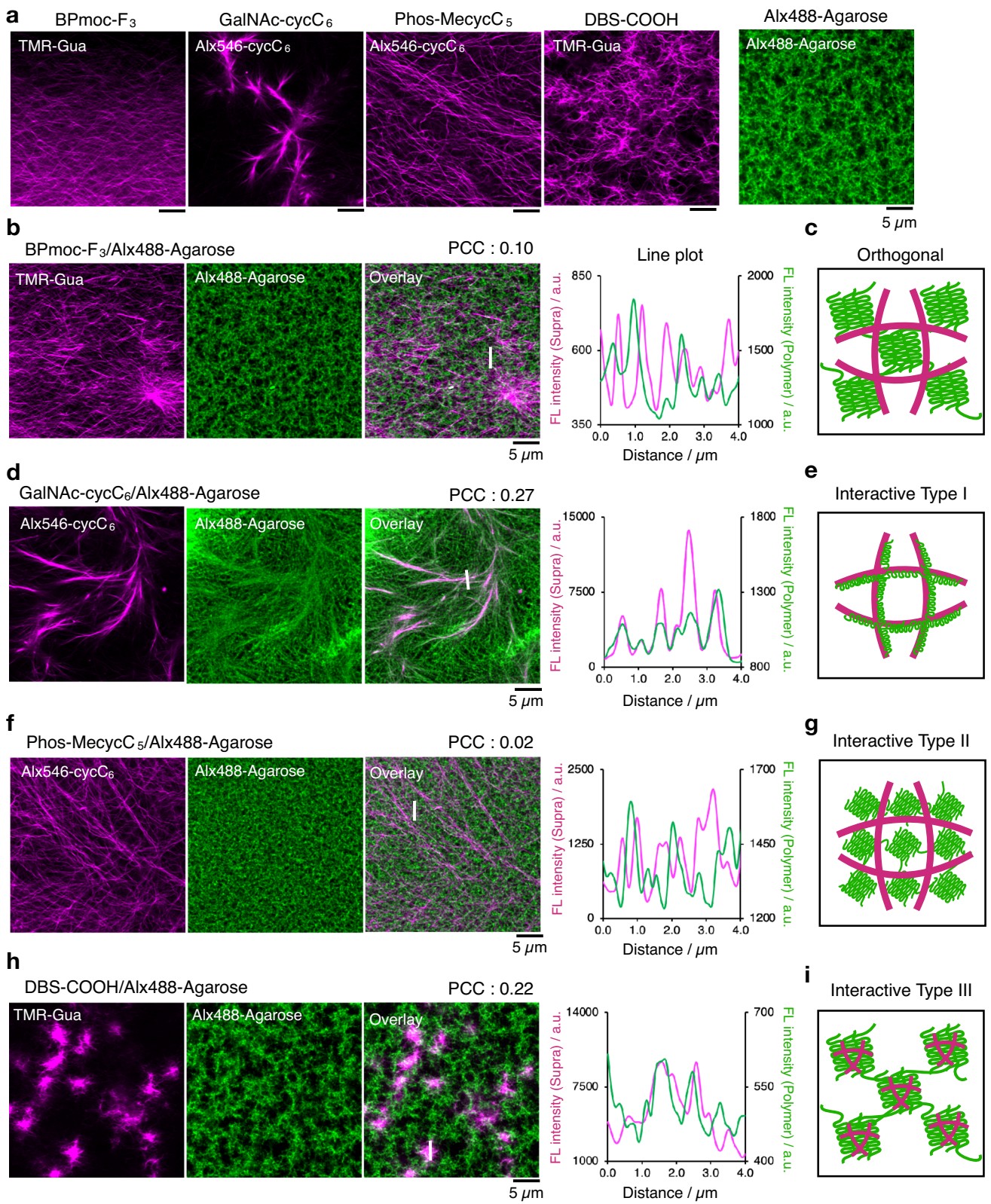

agarose network became more homogeneously dense in the Phos-MecycC₅/agarose composite hydrogel (Supplementary Fig. 4f). The particle analysis quantitatively revealed that the average size of the agarose island domains ($0.211 \pm 0.006\ \mu m^2$) was smaller than the single component agarose ($0.27 \pm 0.02\ \mu m^2$) or the orthogonal BPmoc-F₃/Alx488-Agarose ($0.244 \pm 0.003\ \mu m^2$) (Supplementary Fig. 3b). Alternatively, the morphology of the Phos-MecycC₅ nanofibers remained

nearly identical to the single component (Fig. 2f). The average low Pearson's correlation coefficient ($0.01 \pm 0.02$) and line plot analysis (peak distance: $210 \pm 170$ nm) indicated that the two networks were not correlated with each other (Supplementary Figs. 7 and 9 and Supplementary Table 1). Since the agarose morphology was altered from that of the single-component gel, this network is referred to as interactive network type II (Fig. 2g).

**Fig. 2 | Four distinct network patterns visualized by CLSM imaging. a** CLSM images of single-component hydrogels. **b, d, f, h** CLSM images of **b** BPmoc-F$_3$/Alx488-Agarose, **d** GalNAc-cycC$_6$/Alx488-Agarose, **f** Phos-MecycC$_5$/Alx488-Agarose, and **h** DBS-COOH/Alx488-Agarose. Magenta: supramolecular network, green: agarose network. Line plot analyses along the white lines are shown on the right side of each image. PCC: Pearson's correlation coefficient, FL intensity: fluorescence intensity, Supra: supramolecular network, a.u.: arbitrary units. Scale bar: 5 μm. **c, e, g, i** Schematic illustration of **c** orthogonal, **e** interactive type I, **g** interactive type II, and **i** interactive type III networks. Conditions: [BPmoc-F$_3$] = 0.1 wt% (1.6 mM), [Phos-MecycC$_5$] = 0.4 wt% (6.5 mM), [GalNAc-cycC$_6$] = 0.3 wt% (4.6 mM), [Alx488-Agarose] = 0.5 wt%, [TMR-Gua] = 14 μM, [Alx546-cycC$_6$] = 4.0 μM, [DBS-COOH] = 0.2 wt% (4.49 mM), [glucono-δ-lactone] = 44.9 mM (for DBS-COOH), solvent: 100 mM MES pH 7.0 (for **a, b, d, f**) or water (for **h**), rt. We obtained two more CLSM images of each composite hydrogel and confirmed that the network structures in these images are similar to those shown in Fig. 2. Thus, the CLSM images shown here are the representative ones. The additional images are shown in Supplementary Fig. 12–14. All of CLSM images of hydrogels were acquired in the hydrated state.

CLSM imaging of a composite hydrogel comprising agarose and DBS-COOH was also performed[33,34]. According to Smith's protocol, the composite hydrogel of DBS-COOH/Alx488-Agarose containing TMR-Gua was prepared by reducing the pH via hydrolysis of glucono-δ-lactone (termed a pH decrease protocol, the time course of pH change was shown in Supplementary Fig. 10). In a single-component DBS-COOH hydrogel, DBS-COOH self-assembled into an entangled fibrous structure with a diameter of *ca.* 200 nm that is similar to the peptide- and lipid-type hydrogelators (Fig. 2a). Unexpectedly, in the DBS-COOH/Alx488-Agarose composite hydrogel, DBS-COOH formed heterogeneously distributed dense aggregates with a diameter of 5 μm comprising thinner and shorter nanofibers (Fig. 2h). On the other hand, the agarose showed the sea–island network, nearly identical to those of the single component as confirmed by the histogram analysis [$s = (1.37 \pm 0.03) \times 10^4$] and the particle analysis (domain size: $0.245 \pm 0.003 \, \mu m^2$, pore size: $0.50 \pm 0.12 \, \mu m^2$) (Supplementary Fig. 3 and 4). The average PCC value ($0.23 \pm 0.01$) and line plot analyses (peak top distance: $150 \pm 170$ nm) indicate that the core of the DBS-COOH aggregates were overlapped with the agarose network, and the thinner DBS-COOH fibers at the periphery of the aggregates were in the void space of the agarose (Supplementary Fig. 8, 9 and Supplementary Table 1). This behavior is different from the above-mentioned network patterns, and the morphology of the supramolecular fibers network significantly changed with negligible change of the agarose network. This is referred to as interactive network type III (Fig. 2i).

The CLSM imaging results indicate that there are at least four distinct network patterns in the supramolecular/agarose composite hydrogels. As summarized in Supplementary Fig. 11 and Supplementary Table 2, composite hydrogels of agarose and other LMW gelators were also categorized into the four network patterns based on the PCC value, the standard deviation value, and the islands/void sizes (See Methods and Supplementary Table 3 for details).

## Multiscale imaging reveals hierarchical structures of composite hydrogels comprising multiple fibers

SEM imaging was used to further evaluate the structures of the composite hydrogels. Single-component gels of agarose, BPmoc-F$_3$, Phos-MecycC$_5$, and GalNAc-cycC$_6$ were lyophilized and analyzed in the dried state with SEM, revealing that nanofibers with a diameter of 20–150 nm were bundled to construct the network structures for all the four cases (Fig. 3a). In the freeze-dried BPmoc-F$_3$/agarose composite hydrogel, the SEM images showed many fibers with a diameter of 20–100 nm (Fig. 3b, left). Because there were little differences in the morphology, it was difficult to distinguish between the supramolecular fibers and agarose fibers. In the freeze-dried Phos-MecycC$_5$/agarose and freeze-dried GalNAc-cycC$_6$/agarose gels, indistinguishable nanofibrous structures, such as the BPmoc-F$_3$/agarose gel, were also observed (Fig. 3c, d). The SEM imaging clearly illustrated individual fibers at the nanometer scale, whereas CLSM may visualize the bundled and/or aggregated structures of the nanofibers with sub-micrometer resolution and their network/spatial distribution at the several tens/hundreds micrometer scale (Fig. 3b, middle, right, 3e). In the case of the DBS-COOH/agarose composite hydrogel (interactive type III), Smith's group reported the orthogonal network of DBS-COOH and agarose fibers with SEM images at several tens of nm resolution[34]. The currently conducted CLSM imaging visualized the colocalization of the aggregated DBS-COOH and the island of the agarose network at sub-micrometer resolution. The combined imaging data suggested the hierarchically organized network structure of this composite hydrogel in the range from nanometer to micro/sub-millimeter scale: colocalized micrometer-scale DBS-COOH/agarose network comprising the DBS-COOH and agarose fibers orthogonally formed at nanometer scale. This example may highlight that the use of different microscopy techniques could produce more comprehensive structural information, allowing for in-depth analyses of the multicomponent composite hydrogels.

## Rheological properties of composite hydrogels with four distinct network patterns

We investigated the viscoelastic properties of these four representative composite hydrogels with distinct network patterns. In linear dynamic mechanical tests, the storage modulus ($G'$) for all the composite gels was appreciably higher than the loss modulus ($G''$), and both $G'$ and $G''$ were almost constant in a frequency range between 0.1 and 10 rad s$^{-1}$, confirming they are hydrogels without flowability (Supplementary Fig. 15 and Supplementary Table 4). We noticed that GalNAc-cycC$_6$/Agarose showed a higher $G'$ value (6448 Pa) than the other composite hydrogels (1240–1749 Pa). To quantitatively assess the effect of mixing different gels on stiffness, we defined an enhancement factor as the ratio of the $G'$ value for the composite hydrogel over the sum of $G'$ values for single component gels. To our surprise, the enhancement factor of GalNAc-cycC$_6$/Agarose was 5.4 while those of the other composite gels were below 2.5, suggesting the stiffness of the GalNAc-cycC$_6$/Agarose gel is synergically increased (Supplementary Fig. 15k). The nonlinear viscoelastic properties of the composite gels were also examined to determine the yield strain, defined as the $G'/G''$ cross-over point, in the amplitude sweep experiment (Supplementary Fig. 16 and Supplementary Table 5). The yield strain of GalNAc-cycC$_6$/Agarose (11.6%) was lower than those of any other composite gels (42.0–82.1%). Taken together, the GalNAc-cycC$_6$/Agarose gel is stiffest and most brittle among the composite hydrogels we tested, presumably due to formation of denser crosslinks and/or thicker and less flexible bundles between two different networks. Our results suggest the network patterns of the composite hydrogels might give impacts on their rheological properties.

## Main factors controlling the network patterns of the composite hydrogels

To address the network pattern formation mechanisms, the formation process of the composite hydrogels was investigated using in situ real-time CLSM imaging. A heated solution of the LMW gelator, Alx488-Agarose, and the corresponding fluorescent probe was transferred to a glass bottom dish, after which time-lapse CLSM imaging was immediately started.

In the BPmoc-F$_3$/Alx488-Agarose composite hydrogel (the orthogonal network), the diffusing domains of Alx488-Agarose were observed initially and these were connected with each other to construct the immobilized network within 5 min (Fig. 4a, Supplementary Fig. 17, and Supplementary Movie 1). On the other hand, the assembled structures of BPmoc-F$_3$ could not be clearly identified in the initial

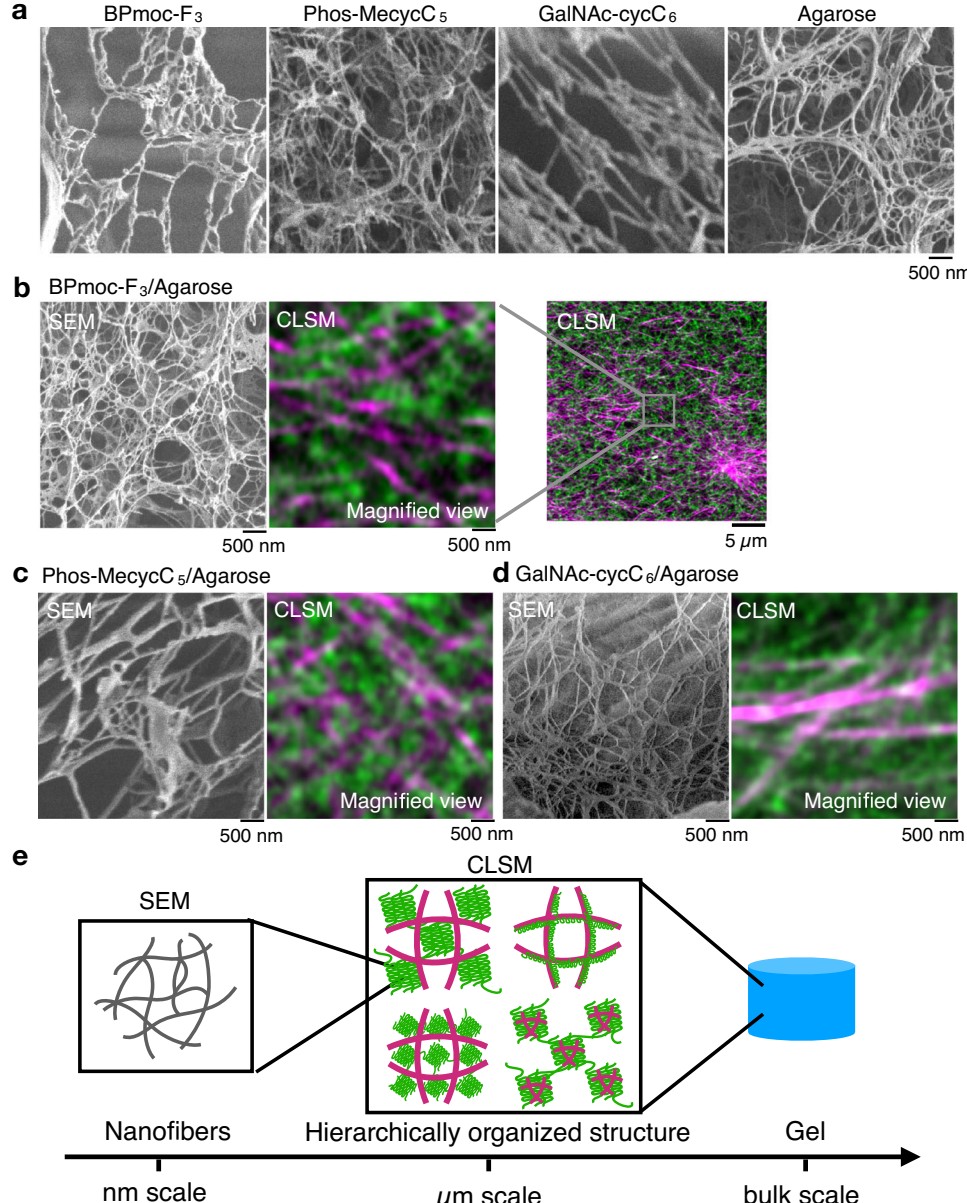

**Fig. 3 | Hierarchical structures revealed by multiscale imaging. a** SEM images of single-component hydrogels. **b** (Left) SEM images and (middle and right) CLSM images of BPmoc-F$_3$/agarose. The middle was a magnified image of the right CLSM image (the same as Fig. 2b). **c**, **d** (Left) SEM images and (right) CLSM images of **c** Phos-MecycC$_5$/agarose and **d** GalNAc-cycC$_6$/agarose. Magenta: supramolecular network, green: agarose network. Scale bar: 500 nm or 5 µm. **e** Schematic illustration of hierarchical structures of the composite hydrogels. Conditions: [BPmoc-F$_3$] = 0.1 wt% (1.6 mM), [Phos-MecycC$_5$] = 0.4 wt% (6.5 mM), [GalNAc-cycC$_6$] = 0.3 wt% (4.6 mM), [agarose] = 0.5 wt%. The freeze-dried samples were used for SEM imaging.

stage. After a lag time of 5–6 min, the short nanofibers of BPmoc-F$_3$ stochastically appeared and elongated in the void space of the agarose network. The difference in formation kinetics was supported by quantitative fluorescence analysis (see Supplementary Methods for details). The number of the agarose domains gradually increased and reached a constant within 5 min (Fig. 4b, green). The time course of the BPmoc-F$_3$ nanofiber formation showed a sigmoidal shaped curve, where the fiber was nearly nothing until the initial 5 min and drastically increased and saturated at *ca*. 13 min (Fig. 4b, magenta). These observations indicated faster formation kinetics of the agarose network than that of the BPmoc-F$_3$ nanofibers.

In contrast, the supramolecular nanofibers formed faster than the agarose network in the Phos-MecycC$_5$/Alx488-Agarose composite gel (the interactive network type II) (Fig. 4c, Supplementary Fig. 18, and Supplementary Movie 2). At the beginning of the time-lapse imaging,

the Phos-MecycC$_5$ nanofibers were already elongated, while the aggregated structure of Alx488-Agarose was not yet observed. After 2 min, a sea–island network with a smaller void space was gradually formed, further developing for *ca*. 6 min. These observations were verified by the time–profile analyses (Fig. 4d). These two examples suggest that the order of the network formation may be a factor that determines the network patterns.

In the GalNA-cycC$_6$/Alx488-Agarose composite hydrogel (the interactive network type I), the order of the network formation is the same as Phos-MecycC$_5$/Alx488-Agarose; the GalNA-cycC$_6$ nanofibers were clearly observed at the beginning, followed by the agarose network formation (Fig. 4e, f, Supplementary Fig. 19, and Supplementary Movie 3). However, in this case, the fibrous-shaped fluorescence of Alx488-Agarose gradually appeared preferentially along the GalNA-cycC$_6$ nanofibers, resulting in the formation of interactive network

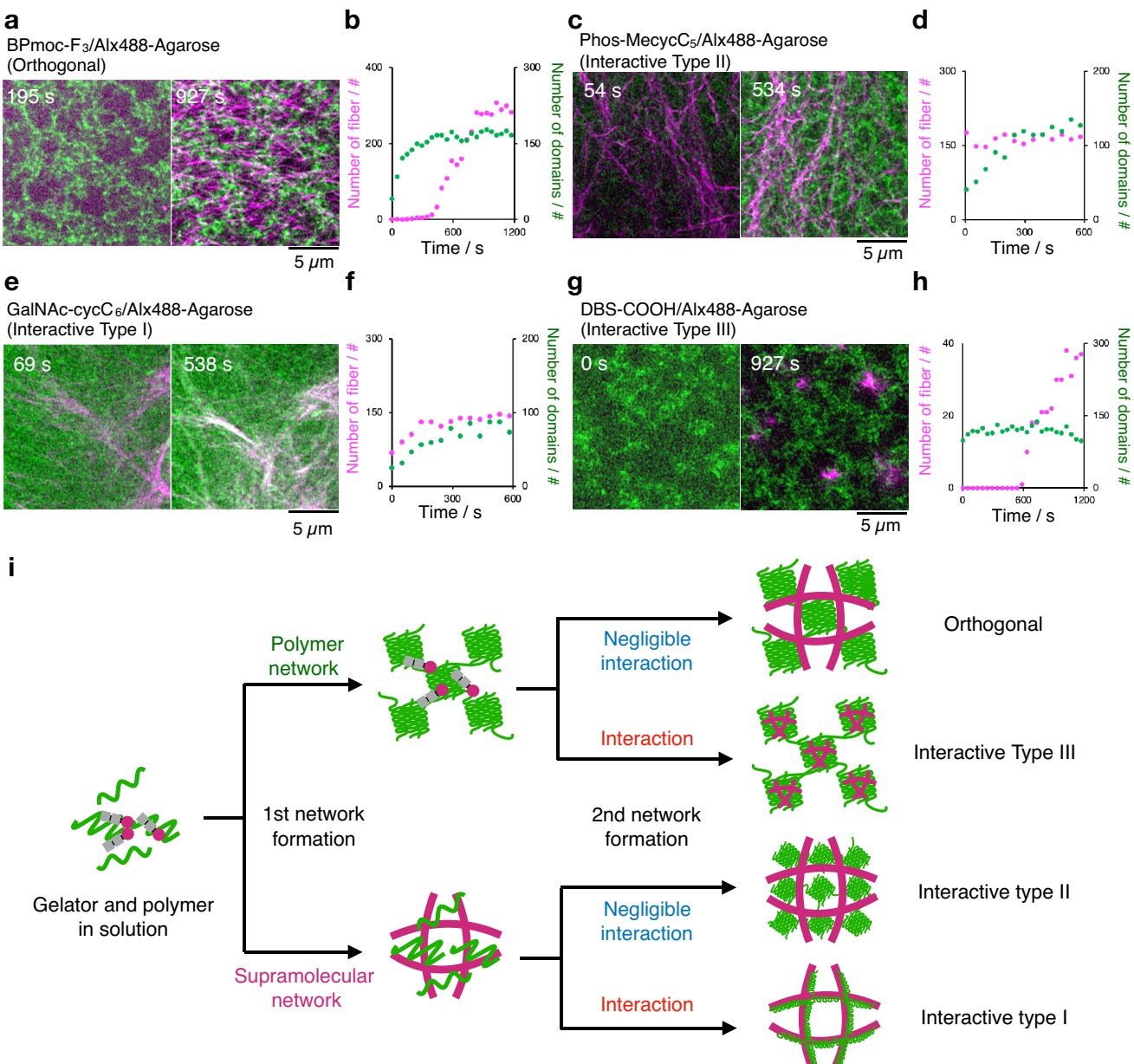

**Fig. 4 | Factors controlling the network pattern formation revealed by real-time CLSM imaging.** Time-lapse images of **a** BPmoc-F₃/Alx488-Agarose, **c** Phos-MecycC₅/Alx488-Agarose, **e** GalNAc-cycC₆/Alx488-Agarose, and **g** DBS-COOH/Alx488-Agarose. Magenta: supramolecular network, green: agarose network. **b**, **d**, **f**, **h** The quantitative analysis of the network formation. Magenta: the number of supramolecular nanofibers counted by the particle analysis, green: the number of the aggregated agarose domain. **i** Proposed mechanism of the network pattern formation. Conditions: [BPmoc-F₃] = 0.1 wt% (1.6 mM), [Phos-MecycC₅] = 0.4 wt% (6.5 mM), [GalNAc-cycC₆] = 0.3 wt% (4.6 mM), [Alx488-Agarose] = 0.5 wt%, [TMR-Gua] = 14 μM, [Alx546-cycC₆] = 4.0 μM, [DBS-COOH] = 0.2 wt% (4.49 mM), [glucono-δ-lactone] = 44.9 mM (for **g**, **h**), solvent: 100 mM MES pH 7.0 (for **a**–**f**) or water (for **g**, **h**), rt.

type I. The time-dependent network formation profile revealed that the GalNAc-cycC₆ nanofibers could work as a template for the agarose network formation via interfiber interactions.

The in situ time-lapse imaging of the DBS-COOH/Alx488-Agarose composite gel (the interactive network type III) was initiated immediately after the addition of glucono-δ-lactone to the hydrogelator solution at approximately rt. The CLSM imaging showed that the sea−island agarose network first formed, followed by gradual formation of the DBS-COOH network (Fig. 4g, h, Supplementary Fig. 20, and Supplementary Movie 4). At 10 min, the distorted spherical puncta of DBS-COOH with a diameter of 1–2 μm emerged inside the dense island region of the agarose network, and then thin and short supramolecular nanofibers grew from these puncta within and near the island domain of the agarose network. The agarose network may assist the formation

of the DBS-COOH fibers, likely through interactions between DBS-COOH and agarose. Therefore, interactions between two fibers/networks may be an additional controlling factor.

On the basis of the network formation observed with in situ CLSM imaging, we propose that the network patterns are governed mainly by two factors: (i) the order of the network formation and (ii) the interactions between supramolecular gelators/fibers and agarose polymers, as shown in Fig. 4i. When the polymer network forms earlier than supramolecular fibers, the self-assembly process of LMW gelators is affected by the presence of the polymer network. If interactions between the gelators and agarose polymers are negligible, the LMW gelators can diffuse and self-assemble independently in the void space of the polymer network, resulting in an orthogonal network. If the LMW gelators and the agarose interact with each other, the

supramolecular fiber formation preferentially occurs in the island domains of the agarose network to afford the interactive network type III. For BPmoc-F$_3$/Alx488-Agarose and DBS-COOH/Alx488-Agarose hydrogels, hydrophobicity of the supramolecular hydrogelators may be important for the interactions. In BPmoc-F$_3$/Alx488-Agarose (orthogonal), BPmoc-F$_3$ has a negative charge due to deprotonation of carboxylic acid at neutral pH, resulting in decreasing hydrophobic interaction with the agarose network. On the other hand, in DBS-COOH/Alx488-Agarose (interactive type III), carboxylate groups of DBS-COOH are protonated at acidic conditions induced by hydrolysis of glucono-δ-lactone, which may enhance interaction between the agarose network probably through hydrophobic interaction and/or hydrogen bonding. Thus, protonated DBS-COOH can nucleate near/inside the agarose network to form the interactive type III network. In the case that the formation of the supramolecular nanofibers is faster than that of the polymer network, the supramolecular network impacts the formation process of the polymer network. When interactions between the supramolecular fibers and agarose are minimal, the polymer diffusion is physically suppressed by the supramolecular network, leading to a homogeneous agarose network with smaller void space (interactive network type II). A similar phenomenon was reported in an interpenetrated hydrogel of agarose with highly charged polyelectrolytes such as xanthan[53]. On the other hand, substantial interactions between agarose and LMW gelators/fibers cause preferential aggregation of the agarose polymers along the supramolecular nanofibers to produce interactive network type I. For GalNAC-cycC$_6$/Alx488-Agarose and Phos-MecycC$_5$/Alx488-Agarose, the surface structure of supramolecular fibers may determine the composite network structure. In GalNAC-cycC$_6$/Alx488-Agarose (interactive type I), GalNAC-cycC$_6$ fibers present lots of sugar moiety on their surface that can interact with agarose polymers through polyvalent sugar-sugar interaction, resulting in interactive type I. In Phos-MecycC$_5$/Alx488-Agarose (interactive type II), Phos-MecycC$_5$ forms polyanionic supramolecular nanofibers due to a negative phosphate group, which may cause a negligible interaction with agarose polymers. We also confirmed no significant differences in the formation kinetics of supramolecular fibers and agarose network between the single-component and composite hydrogels in all cases, implying that the interaction between supramolecular gelators/fibers and agarose is not strong enough to alter the formation kinetics (Supplementary Fig. 21).

The formation processes for composite hydrogels of agarose and other LMW gelators agree well with the proposed mechanism (Supplementary Fig. 22–24). Additionally, in some cases, the network patterns depend on the concentration of LMW gelators. For example, the network pattern of BPmoc-F$_3$/Alx488-Agarose changed from the orthogonal to the interactive type II when the concentration of BPmoc-F$_3$ was increased from 0.1 wt% to 0.4 wt% (PCC values: $0.08 \pm 0.10$ and $0.04 \pm 0.13$, respectively; Supplementary Fig. 25). Time-lapse CLSM imaging of the composite hydrogel containing 0.4 wt% BPmoc-F$_3$ revealed the order of the network formation was reversed; specifically, the nanofiber formation of BPmoc-F$_3$ was faster than the agarose network formation (Supplementary Fig. 26). This order change explained the alteration of the network pattern from orthogonal to interactive type II (Fig. 4i). These results also support our proposed mechanism of the network pattern formation.

We also investigated whether the network patterns depend on the protocol of hydrogel preparation. A BPmoc-F$_3$/Alx488-Agarose composite hydrogel was prepared with the pH decrease protocol. CLSM imaging of this composite gel showed characteristics of both orthogonal and interactive type III networks. In this case, the agarose network maintained its original morphology, and BPmoc-F$_3$ formed both well-elongated nanofibers and spherical aggregates (Supplementary Fig. 27a and 27b). Most BPmoc-F$_3$ nanofibers localized independently of the agarose domains and the spherical aggregates merged with the agarose domains, which were similar to the orthogonal and interactive

III networks, respectively. Time-lapse imaging of the formation process showed the agarose network formed earlier than supramolecular aggregates and fibers (Supplementary Fig. 27c, d, Supplementary Movie 5). The spherical puncta of BPmoc-F$_3$ initially emerged inside the island region of the agarose network, and then some of the fibers elongated from the puncta towards both the inside and outside of the agarose network and others generated and grew at void spaces of the agarose network (Supplementary Fig. 27c). This composite hydrogel thus has the features of both the interactive type III and the orthogonal networks. These results indicate that the network patterns may be controlled by the above-mentioned two factors, the order of network formation and interactions between supramolecular gelators/fibers and agarose polymers, while these factors can be affected by the protocol of hydrogel preparation. Furthermore, this finding highlights that the composite gel networks are not always definitely categorized into four patterns and some may be characterized as an intermediate between the four patterns.

## Fracture-induced remodeling of a composite network using dynamic properties

During this study, we noticed that only composite hydrogel of agarose and NPmoc-F(F)F spontaneously changed its network at the several hundred micrometer scale during the prolonged incubation. The network structure of the as-prepared NPmoc-F(F)F/Alx488-Agarose composite hydrogel was classified into interactive type II (Fig. 5a, Supplementary Fig. 11b). However, after 24 h incubation, many of the highly fluorescent NPmoc-F(F)F domains of *ca.* 500 μm diameter appeared with heterogeneous distribution, whereas the fluorescent intensity of the other regions substantially decreased (Fig. 5b, left). The agarose network also altered the heterogeneous distribution, where the darker regions spatially corresponded to the brighter NPmoc-F(F)F domains, as confirmed by the line plot analysis and Pearson's correlation coefficient (before and after incubation: 0.29 and −0.42, respectively) (Fig. 5b, middle, right). Such structural change of the NPmoc-F(F)F/Alx488-Agarose composite hydrogel seems similar to phase-separation processes like an aqueous two-phase system[54–62]. Notably, we found that other composite hydrogels investigated in this study did not show any time-dependent macroscopic network change for at least 48 h (Supplementary Figs. 28 and 29). To understand why the unique phase separation occurred only in the NPmoc-F(F)F/agarose composite gel, we investigated the packing structure of the NPmoc-F(F)F nanofibers before and after the phase separation by CD spectrometry (Supplementary Fig. 30). The CD measurement of the NPmoc-F(F)F/Alx488-Agarose hydrogel showed the Cotton peaks derived from the NPmoc-F(F)F nanofibers were completely different before and after the incubation (agarose did not show any Cotton peaks under this measurement condition[40]) These results suggest that the as-prepared NPmoc-F(F)F nanofibers are in the metastable state and reassemble into a more thermodynamically stable state during aging, which may be one of the key factors to induce the phase-separation behavior.

Such properties in the dynamic network change may allow for artificial remodeling of this composite hydrogel network. We initially tested if the network change of the NPmoc-F(F)F/Alx488-Agarose was triggered when a pinhole was created by puncturing with a 100-μm diameter needle (Fig. 5c). Immediately after the puncture of the composite hydrogel, a dark circle with a diameter of *ca.* 100 μm was observed in both NPmoc-F(F)F and agarose channels (Fig. 5d, left). During incubation, the fluorescence intensity of TMR-Gua (corresponding to NPmoc-F(F)F fibers) in the pinhole area gradually increased for 4 h, while the intensity in the non-punctured area decreased such that the fluorescence intensity in the pinhole area became much greater than that in the non-punctured region (Fig. 5d, e, Supplementary Fig. 31). This result suggests that the network remodeling may be controlled by diffusion of the NPmoc-F(F)F gelator from

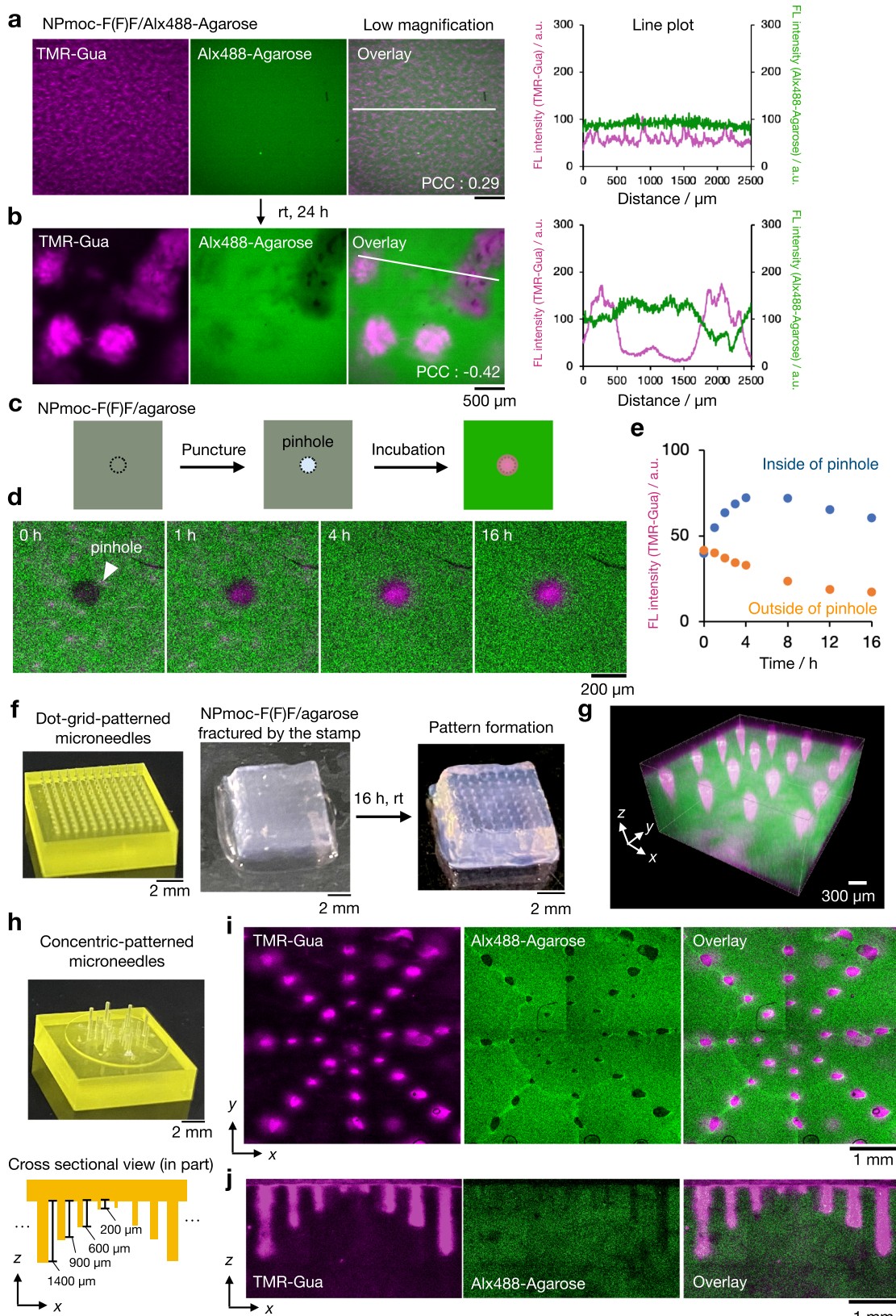

the outside of the pinhole. In contrast, the fluorescence intensity of Alx488-Agarose showed negligible changes in and around the pinhole (Supplementary Fig. 31c). Additionally, the response in dynamic network remodeling depends on the size of the pinhole. For the larger pinhole (*ca.* 500 μm), the NPmoc-F(F)F nanofibers accumulated along the outer edge of the pinhole, not inside the pinhole, to form a ring-

shaped pattern with a width of *ca.* 400 μm (Supplementary Fig. 32). In contrast, the fluorescence intensity of the Alx488-Agarose network at the outer edge of the pinhole significantly decreased, suggesting that Alx488-Agarose was extruded from the increasing area of the NPmoc-F(F)F fibers (Supplementary Figs. 32b and 32c). These results indicate the fracture-induced condensation of the supramolecular NPmoc-F(F)

**Fig. 5 | Fracture-induced spatial patterns of the NPmoc-F(F)F/agarose composite hydrogel. a, b** Low-magnified CLSM images of NPmoc-F(F)F/Alx488-Agarose composite hydrogel: **a** before and **b** after incubation for 24 h. Line plot analyses along the white lines are shown on the right side of each image. PCC: Pearson's correlation coefficient, FL intensity: fluorescence intensity, a.u.: arbitrary units. **c** Schematic illustration of fracture-induced network remodeling. **d** Time-lapse CLSM imaging of the punctured NPmoc-F(F)F/Alx488-Agarose composite hydrogel. **e** Time-course change of the FL intensity of TMR-Gua inside and outside the pinhole. Regions of interest were shown in Supplementary Fig. 31a. **f** Fracture-induced spatial gel patterning with designer 3D printed microneedle stamps. (Left)

A photograph of a microneedle stamp with a dot–grid pattern and a composite hydrogel (middle) immediately and (right) 16 h after punctuation. **g** 3D CLSM images of the patterned hydrogel. **h** (Top) A photograph of a microneedle stamp with a concentric pattern. (Bottom) An illustration of the side view of the stamp. **i** CLSM images and **j** xz-sectional images of the hydrogel patterned by the concentric-patterned stamp. Magenta: supramolecular network, green: agarose network. Conditions: [NPmoc-F(F)F] = 0.4 wt% (7.0 mM), [Alx488-Agarose] = 0.5 wt%, [TMR-Gua] = 14 μM, solvent: 100 mM MES pH 7.0, rt. Scale bar: 500 μm **a**, **b**, 200 μm **d**, 2 mm **f**, **h**, 300 μm **g**, and 1 mm **i**, **j**.

F nanofiber network around the pinhole produced a new phase-separated structure. Notably, such network remodeling was not observed in the 24 h aged NPmoc-F(F)F/Alx488-Agarose composite hydrogel and the other composite hydrogels with the interactive network type II (Supplementary Figs. 33 and 34). These results again supported that the dynamic feature of the metastable as-prepared NPmoc-F(F)F/Alx488-Agarose, rather than the network pattern, is critical for the fracture-induced condensation of the NPmoc-F(F)F network[51].

We created spatially controlled macroscopic patterns of the composite hydrogel by puncturing with microneedle stamps fabricated by 3D printing. The NPmoc-F(F)F/Alx488-Agarose composite hydrogel was punctured by a microneedle stamp with a dot–grid–pattern (width: 300 μm, height: 900 μm, interval: 700 μm, Fig. 5f, left). After incubation for 16 h, the macroscopic dot–grid pattern of opaque areas was observed on the surface of the hydrogel (Fig. 5f, right). CLSM imaging visualized that the NPmoc-F(F)F network accumulated in the punctured areas and the condensed regions were generated in the plane and along a depth comparable to the needle length, resulting in the formation of the 3D dot–grid phase-separated patterns (Fig. 5g, Supplementary Fig. 35). By use of a concentric-patterned microneedle stamp with varied needle lengths of 200 μm to 1400 μm, we also succeeded in fabricating macroscopic patterns with a controlled depth profile of the condensed region of the NPmoc-F(F)F nanofibers (Fig. 5h–j, Supplementary Fig. 36). Such fracture-induced remodeling combined with the micrometer-precision 3D printing technique may provide a new user-friendly method for spatially programmable patterning of functional hydrogels in two and three dimensions.

## Discussion

The designed composite hydrogels comprising supramolecular nanofibers and covalent polymers can have four distinct network structures governed by two key factors, the order of network formation and interfiber interactions. Furthermore, 3D spatial patterning within the composite gel was achieved at the scale from 100 μm to >1 mm via fracture-induced network remodeling using designer 3D printed microneedle stamps. To date, the similar spontaneous phase separation behaviors have been reported for multicomponent hydrogels comprising two distinct supramolecular fibers[60–62], whereas we demonstrated spontaneous and fracture-induced phase separation with a combination of supramolecular fibers and covalent polymers in the field of supramolecular and polymer chemistry. Such synthetic composite materials with ECM-inspired dynamic, programmable hierarchical networks exhibit unique mechanical properties and life-like adaptivity because of synergistic effects among distinct networks, which are promising for various applications such as controlled drug release, tissue engineering, and regenerative medicine.

## Methods
### CLSM imaging of composite hydrogels prepared by a heat–cool protocol
A suspension of a gelator powder and agarose in 100 mM MES (pH 7.0) was created and 1% vol TMR-Gua [1.4 mM, 100 mM MES pH 7.0 (10%

DMSO)] and/or Alx546-cycC$_6$ (400 μM, 100 mM MES pH 7.0) was added. The suspension was dissolved using a heat gun (PJ-206A1, Ishizaki Electric Mfg. Co., Ltd, Japan). After cooling to rt, the resultant mixture (20 μL) was transferred to a glass bottom dish (D11530H, Matsunami Glass Ind., Ltd., Japan). After incubation at rt for 2 h in the presence of water to prevent the hydrogel drying out, CLSM imaging was conducted. The assay conditions are referred to in the figure captions.

### CLSM imaging of composite hydrogels prepared by a protocol with decreasing pH
An aqueous suspension of gelator and agarose was created and 1% vol TMR-Gua [1.4 mM, 100 mM MES pH 7.0 (10% DMSO)] and 5% vol NaOH aq. (1.0 M) were added. The suspension was dissolved using a heat gun. After cooling to rt, the resultant (100 μL) was mixed with glucono-δ-lactone (10 μL, 449 mM, water) and immediately transferred to a glass bottom dish. After incubation at rt for 12 h in the presence of water to prevent the hydrogel from drying out, CLSM imaging was conducted. The assay conditions are referred to in the figure captions.

### Scanning electron microscopy
Hydrogels were frozen by immersing in liquid nitrogen and lyophilizing overnight. The samples were placed on a conductive carbon adhesive tape (thin aluminum foil core) and sputter coated with a thin layer of Au (*ca*. 5 nm). Secondary electron images were acquired using a field emission scanning electron microscope (SU8200, Hitachi High-Tech Cooperation, Japan) at 1.0 kV voltage.

### Real-time CLSM imaging of the network formation
The suspension of a gelator and Alx488-Agarose in 100 mM MES, pH 7.0 was dissolved using a heat gun. The resultant mixture (20 μL) was immediately transferred to a heated glass bottom dish. After a couple of minutes, CLSM imaging was conducted. For the DBS-COOH/Alx488-Agarose composite hydrogel, 1% vol TMR-Gua [1.4 mM, 100 mM MES pH 7.0 (10% DMSO)] and 5% vol NaOH aq. (1.0 M) was added to an aqueous suspension of DBS-COOH powder and Alx488-Agarose. The suspension was dissolved using a heat gun. After cooling to rt, the resultant solution (100 μL) was mixed with glucono-δ-lactone (10 μL, 449 mM, water) and immediately transferred to a glass bottom dish to start time-lapse CLSM imaging.

### Fracture-induced patterning of the NPmoc-F(F)F/agarose network
A suspension of NPmoc-F(F)F (0.4 wt%, 7.9 mM) and agarose (0.5 wt%) in 100 mM MES (pH 7.0) was created and 1% vol TMR-Gua [1.4 mM, 100 mM MES pH 7.0 (10% DMSO)] was added. The suspension was dissolved using a heat gun. After cooling to rt, the resultant mixture (200 μL) was transferred to a rectangular PDMS mold (*ca*. 7.0 mm × 7.0 mm × 3.5 mm). After 2 h, fracture-induced patterning of the hydrogel was created using a plunger for nanoliter injection (World Precision Instruments, USA) for a larger pinhole (diameter: *ca*. 500 μm), microneedles fabricated by a 3D printer (microArch®S140, Boston Micro Fabrication, USA) for a smaller pinhole (diameter: *ca*. 100 μm) and macroscopic patterns.

## Data availability

The relevant experimental data that support the findings of this study have been deposited as the Source Data file. The other data generated during this study are available in numerical format from the corresponding author upon request. Source data are provided in this paper.

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

## Acknowledgements

We thank Ashleigh Cooper, PhD, from Edanz (https://jp.edanz.com/ac) for editing a draft of this manuscript. We thank Boston Micro Fabrication for providing a patterned microneedles fabricated by a 3D printer. This work was supported by a Grant-in-Aid for Scientific Research on Innovative Areas "Chemistry for Multimolecular Crowding Biosystems" (JSPS KAKENHI Grant Number JP17H06348), the Japan Science and Technology Agency (JST) ERATO Grant Number JPMJER1802 to I.H., and by a Grant-in-Aid for Young Scientists (JSPS KAKENHI Grant Number JP20K15400), a Grant-in-Aid for Scientific Research (B) (JSPS KAKENHI Grant Number JP22H02195) to R.K., and by JST Spring, Grant Number JPMJSP2110 to K.N.

## Author contributions

I.H., R.K., and K.N. conceived and designed the project. K.N. conducted all experiments. R.K. and K.N. conducted SEM experiments. K.N., T.A., and K.U. conducted rheological experiments. I.H., R.K., and K.N. analyzed the data and wrote the manuscript. All authors discussed and commented on the manuscript.

## Competing interests

The authors declare no competing interests.
