## [Peer Review File · Nature Communications]

Four distinct network patterns of supramolecular/polymer composite hydrogels controlled by formation kinetics and interfiber interactionsReviewers' Comments:

Reviewer #1:

Remarks to the Author:

This is an interesting manuscript that uses confocal laser scanning microscopy to gain detailed insight into the assembly of hybrid hydrogels based on supramolecular gelators with an agarose polymer gel network. This technique has enabled the authors to achieve unique discrimination between different assembly modes. In particular, they are able to probe the co-localisation, or spatially-separated assembly of different combinations of gelators. Of particular value is that the authors have used a variety of different gelators, which has enabled them to characterise a range of different co-assembly models. This moves the field of hybrid hydrogels forwards quite significantly, and in my view, the paper would be suitable for publication in Nature Communications with some modifications.

It is worth noting that the supramolecular gels in this work are assembled by slightly different mechanisms. Most of them are formed using a heat-cool cycle, while one of them is assembled via slow acidification. The authors do not comment on whether assembly mode would impact on the morphologies that result. Peptide-acid gelators can often also be assembled by slow acidification. I wonder if this had been done, whether it would have changed the observed morphology compared with the system formed via a heat-cool cycle. Presumably it may change the kinetics of assembly, and also the order in which the networks form - this may impact on final morphology. The requirement of the acid to trigger gelation in one of the cases may also modify its assembly mode - maybe this process preferentially occurs within the preformed agarose 'sea islands' either because the acid accumulates there, or the effective pH is somewhat different within those domains.

In terms of the 'sea islands' in the agarose gel, it can sometimes be quite difficult to see these from the imaging alone (although in other cases it is more clear). I'm not sure I would have spotted them if they hadn't been pointed out. There is some statistical analysis in the supporting information which helped with this in distinguishing between larger and smaller void spaces.. I did wonder whether early on in the paper, it would be worth the authors providing some quantification in the paper itself of how they assessed the extent of sea-island formation in agarose alone, so that then they can benchmark this against the hybrid gels they report.

I really like the quantification of overlap and non-overlap between different gel networks that can be achieved using the CLSM technique - it is extremely powerful. The differences in behaviour between some of the supramolecular gelators is really clear. The authors also do a good job with the time-resolved studies in terms of rationalising many of their observations based on the order in which the networks form.

The authors contrast their observations of DBS-COOH assembly with those of Smith et al. It is perhaps worth noting that in reference 34, these researchers only noted that separate fibres of DBS-COOH and agarose could be observed by SEM - which they thus defined as 'orthogonal'. They did not comment on whether the networks were co-localised within the wider sample, or were fully homogeneously distributed. Indeed, this would be very difficult to ascertain using their method of SEM with freeze drying. As such, Smith's report of 'orthogonal fibres' for this combination is not necessarily inconsistent with this paper's report that the networks co-localise. The authors could consider the wording on page 12.

The authors report an interesting example in which they can achieve spatial resolution of one of the supramolecular gels. It is evident that over a time period of 24 hours, this system undergoes an effective phase separation, with the supramolecular gelator forming clustered aggregates. Phase separation is not unusual in polymer (gel) blends, being driven by thermodynamics of the material on the bulk scale. Some references to phase separation processes may be appropriate, and using that wording rather than 'segregation' may be better. It is interesting that the supramolecular gel is the one that appears to move rather than the agarose - presumably because the fibres can potentially

assemble/disassemble (or are more flexible)?

Once the materials are punctured, this presumably provides a void space that will encourage the phase separation process. Phase separations always want to minimise surface energies, and the newly formed voids at the puncture points will create high interfacial energies, which can presumably be somewhat satisfied by filling them with the supramolecular gelator – which anyway wants to phase separate – hence directing the phase separation process. I wonder whether this type of patterning should perhaps be described as 'directed phase separation', or possibly 'mechanically-directed phase separation'. I'm not sure whether such processes are well known in polymer blend materials - but it would be worth checking as there may be analogies that could be drawn. Alternatively, it may actually be something that supramolecular systems are much better set up to deliver as a result of their greater dynamics. It is, in any case, a clever way of controlling the assembled microstructure of a hybrid gel system.

In summary, this is an interesting manuscript, which will capture the attention of researchers both within and beyond the immediate field, and I am supportive of publication once the authors have considered my comments.

Reviewer #2:

Remarks to the Author:

In this work, Nakamura et al. investigates different supramolecular hydrogel composites made of agarose and various low molecular weight (LMW) gelators. Using in-situ confocal imaging, the different localization of agarose and LMW gelators were characterized and categorized into four different network patterns. These network patterns were claimed to be governed by the order of network formation and interaction between supramolecular gelators/fibers and agarose. Although the phenomena observed between the different systems is interesting, the current work lacks the discussion to rationalize the observation presented by the authors. Additional studies also need to be done to support the claims of the authors in different parts of the paper. The authors also do not make an attempt to relate the structures of their materials to any sort of physical attributes, which is important for the context and rationale behind this work. This paper is therefore not recommended for publication in Nature Communications. The following comments are listed below to help the authors improve the paper.

1. While the CLSM images show a difference in morphology of the separate network patterns, there is no difference observed between the SEM images. Therefore, the entirety of the work is relying on the given set of CLSM imaging. It would be beneficial to have at least one more method to observe morphological differences between the samples. A possible solution could be to use Cryo-FIB SEM. This can better preserve the morphology of the structure and eliminates drying effects that can be generated when using lyophilization.

2. In the introduction, the authors highlight the importance of different network patterns on the mechanical performance of tissues (tension, compression). Did the authors try to investigate whether there were any mechanical differences for each of the network patterns they claimed to achieve? Are there any rheological investigations performed to see how the networks differ in their viscoelastic properties? Tensile, compressive and rheological testing for each of the different networks would improve the impact of this work.

3. On page 5, when the different LMW gelators were introduced, there were no explanation as to why the specific molecular structures were selected for the study. Discussing it at this point of the paper will help the reader.

4. There are only single line plots presented for each overlay of the CLSM image. Additional line plots should be provided in the supporting information.

5. In general, additional CLSM micrographs in the supporting information will help the reader and reviewers to determine if the presented micrographs in the main text are representative of the sample as a whole.

6. The schematic illustrations of the interactive type I is slightly misleading, as it does not seem clear from the CLSM image that the agarose aligns with the LMW gelator fibers as is drawn in the schematic. Additionally, it is hard to see a clear difference in pattern between orthogonal and interactive type II from the CLSM images, as opposed to what the schematic illustrations suggest.

7. A pH dependence experiment for the DBS-COOH composite is missing.

8. SEM micrographs show images of the gel after lyophilization whereas CLSM are obtained in its hydrated state. This must be clearly distinguished in the paper.

9. Were kinetics studies of the single component hydrogels performed? The only ones presented were that of the supramolecular composites.

10. The authors claim that the formation processes for the composite hydrogels of agarose and LMW gelators is governed by the interactions between the gelator and the agarose. However, there are no discussion on how the different chemical structure of the LMW gelator differed in their interaction with the agarose. A detailed rationale of the interaction between the gelators and fibers, and how they related to the resulting structure of the composite is needed.

11. In addition, the authors also differentiated between the (i) order of network formation and (ii) interaction between the gelators and agarose as the basis for network formation of the composite. However, would the network of formation not be heavily influenced by the interaction between the gelators and agarose as well? The kinetics study of the single component hydrogel would provide good supporting information with this part of the study.

12. Would the authors expect the same fracture-induced remodelling for other supramolecular hydrogel composites tested other than the NPmoc-F(F)/Alx488-Agarose?

13. Page 5, lines 13-16 needs citation and supporting literature.

Reviewer #3:

Remarks to the Author:

The authors provide important new insight into the assembly of hydrogel networks between a polymeric hydrogel and many small molecule hydrogels. Confocal light scanning microscopy is the main tool used for their analysis. The authors distinguish several different morphologies which they ascribe to the kinetics of gel formation and interactivity between the gels. They also explore the dynamic reorganization of one of the gel systems in good detail. The work provides a methodology for the identification of several different hydrogel network types using a simple CLSM setup. Furthermore, they give rationale for how each of the hydrogel networks can be targeted, through the adjustment of the kinetics of the assembly of the hydrogels, as well as the interactivity between the gels. This is a significant advancement in hydrogel network design. The experiments appear to be conducted properly. Experimental detail is sufficient for reproduction, though the number of trials/images used for data collection should be specified.

The goal of the research project is good and the authors have designed some effective systems to achieve their goal of hydrogel network classification and understanding of the mechanisms by which different networks are formed. However, the evidence sometimes does not fully support the claims

and, in some cases, even contradicts their conclusions. At the crux of my concerns is their classification of the various networks. The rationale why they chose to classify some networks as orthogonal, interactive, and the different types of interactive networks is not clearly explained, nor is it entirely clear as to the exact cutoffs for the classification of each type of network. With these concerns, the manuscript needs major revision before acceptable for publication. Some specific comments are summarized below:

Specific comments:

- 1) For BPmoc-F3 and Agarose (Figure 2b) the Pearson's coefficient is 0.1, which indicates weak to no correlation between the two assemblies. However, this is not exactly what is shown in the schematic diagram used to represent this system (Figure 2c). The diagram shows an "Orthogonal" distribution of the two assemblies, which would correspond to a negative Pearson's correlation coefficient. What is the reasoning for this?
- 2) Figure 2e has a faint "G" in the center of the schematic. Either explain its inclusion or remove it if it is an error.
- 3) While there is obvious interactive network formation between GalNAc-cycC6 and Agarose (Figure 2d), the PCC remains quite low for what is shown as a near one-for-one assembly in Figure 2e. Why is the PCC so low in this situation?
- 4) Figure 2f shows a PCC of 0.02, but the text (Page 8, Line 19) indicates a PCC of -0.01. Please correct this discrepancy.
- 5) Page 8, Line 19 "The low Pearson's correlation coefficient (-0.01) and line plot analysis indicated that the two networks were well segregated." A strongly negative PCC would indicate segregation, but a PCC near 0 indicates no correlation.
- 6) The PCC of Interactive Type II network is smaller (-0.01) than that of orthogonal network (0.10). Since this seems to be a large part of the basis for your classifications, why do you ascribe orthogonal characteristic to one and interactive characteristic to the other?
- 7) For Interactive Type II (Figure 2f,g) the claim is made that the void size in the agarose sea-island network decreased so that there is a more uniform distribution when compared to single component agarose, such that the agarose islands are the same size but the distance between islands is smaller. However, the schematics 2c and 2g suggest that the agarose islands are decreasing in size along with the space between the islands getting smaller as well. Could you clarify if the claim is that both the island and void space is getting smaller, or just the void space, and adjust the figures accordingly? If the islands are getting smaller, particle size quantification would be helpful in backing up this claim.
- 8) Page 9, Line 9: "This behavior is different from the above-mentioned network patterns, where the morphology of the supramolecular fibers network significantly changed with negligible change of the agarose network." I believe that this sentence is improperly worded, as it gives the impression that the previous networks show no change in agarose assembly.
- 9) The scale of the images used in the analysis is $\sim 25 \mu\text{m} \times 25 \mu\text{m}$. Since the image analysis is the crux of the manuscript, it is important for the reader to know how many images/how large of an area was surveyed in each case. Please indicate how many images were used in each analysis.
- 10) Page 15, Line 17: "The agarose network may assist the formation of the DBS-COOH fibers, likely through interactions between DBS-COOH and agarose. Therefore, interactions between two fibers/networks may be an additional controlling factor." A control experiment comparing the kinetics of DBS-COOH assembly with the co-assembly would be helpful in assessing the validity of this claim. For the interactive type networks, there should be some rate differences between the co-assembly and lone assemblies of the gelators/agarose.
- 11) If Interactive Network Type II (Figure 4i, Page 16, Line 6) is the result of non-interaction between the agarose and LMW gelator, why it is called interactive?
- 12) The experiment in which BPmoc-F3 concentration is increased to 0.4 wt% shows an obvious and significant change in supramolecular assembly. The authors claim that this is a change from Orthogonal to Interactive Type II network pattern. However, the PCC of the new network is becoming more negative (-0.07) indicating a network that is more orthogonal. This further throws into question the naming convention of the proposed networks, why is this negatively correlated network named Interactive? Is there a further increase in the negative correlation upon further increase of the BPmoc-

F3 concentration?

13) Page 16, Line 19: "Time-lapse CLSM imaging of the composite hydrogel containing 0.4 wt% BPmoc-F3 revealed the order of the network formation was reversed; specifically, the nanofiber formation of BPmoc-F3 was faster than the agarose network formation (Supplementary Fig. 13)." The figure in questions shows that both agarose and supramolecular fibers exist at the start of the assembly process. Based on the frames shown, it appears that the agarose gel is fully formed at 54 s, while the supramolecular fiber is only beginning to form at this point. This appears to go directly against the claims in the body of the manuscript. Is there an explanation for this?

14) The gel remodeling experiments are very interesting, elegant, and thoroughly explored. However, they raise an issue that I have mentioned a few times in my previous comments. Specifically, it appears that Interactive Type II networks are in fact repulsive/exclusionary in nature, but are kinetically trapped in some metastable state that appears slightly orthogonal on the small scale. Upon the fracturing, the gels reassemble into their preferred exclusionary assembly, as evidenced by the many micro-needle experiments and the strongly negative PCC (-0.42). Based on Supplementary Figure 14, the PCC of the NPmoc-F(F)F system is -0.52 when measured at the same scale as the other systems. This suggest to me that this is not an interactive system driven by favourable interactions, but rather an exclusionary assembly driven by repulsion. Is the reorganization behavior common to all Interactive Type II systems or only the NPmoc-F(F)F hydrogelator? Do you see reorganization for the orthogonal networks under similar conditions?

15) The classification of the networks, while ambitious, appears to have several issues which keep it from being a robust classification protocol. The exact method by which the classifications were arrived at is quite vague in the text and remains largely qualitative. If the goal of the manuscript is to provide the reader with a universal method for classifying various hydrogel networks, a more rigorous, step-by-step guide should be provided. Perhaps better classification would be to use Orthogonal, Interactive Type I, Interactive Type II, and Exclusionary (Or Repulsive) to account for the negative PCC observed in some of the networks.

16) Supplementary Figure 14 shows DBS-COOH/Alx488-Agarose as Orthogonal, but the body of the manuscript labels it as Interactive Type III.

Reviewers comments:

Reviewer 1 (Remarks to the Author):

Comment 1:

This is an interesting manuscript that uses confocal laser scanning microscopy to gain detailed insight into the assembly of hybrid hydrogels based on supramolecular gelators with an agarose polymer gel network. This technique has enabled the authors to achieve unique discrimination between different assembly modes. In particular, they are able to probe the co-localisation, or spatially-separated assembly of different combinations of gelators. Of particular value is that the authors have used a variety of different gelators, which has enabled them to characterise a range of different co-assembly models. This moves the field of hybrid hydrogels forwards quite significantly, and in my view, the paper would be suitable for publication in Nature Communications with some modifications.

Reply 1:

We appreciate your careful review and positive comments to our manuscript. To address your concerns, we newly conducted additional experiments and carefully amended the manuscript as shown below. All modifications in the main text and supplementary information are highlighted in red letters.

Comment 2:

It is worth noting that the supramolecular gels in this work are assembled by slightly different mechanisms. Most of them are formed using a heat-cool cycle, while one of them is assembled via slow acidification. The authors do not comment on whether assembly mode would impact on the morphologies that result. Peptide-acid gelators can often also be assembled by slow acidification. I wonder if this had been done, whether it would have changed the observed morphology compared with the system formed via a heat-cool cycle. Presumably it may change the kinetics of assembly, and also the order in which the networks form - this may impact on final morphology. The requirement of the acid to trigger gelation in one of the cases may also modify its assembly mode - maybe this process preferentially occurs within the preformed agarose 'sea islands' either because the acid accumulates there, or the effective pH is somewhat different within those domains.

Reply 2:

We appreciate your important suggestion. According to the reviewer's comment, we prepared a **BPmoc-F₃**/Agarose hydrogel with a pH decrease protocol and conducted CLSM imaging (Supplementary Fig. 27). As a result, this composite gel shows characteristics of both orthogonal and interactive type III. In this case, the agarose network maintained its original morphology, and **BPmoc-F₃** formed both well-elongated nanofibers and spherical aggregates. Most **BPmoc-F₃** nanofibers localized independently of the agarose domains and the spherical aggregates merged with the agarose domains, which were similar to the orthogonal and interactive type III network, respectively. Time-lapse imaging of the formation process showed that the spherical puncta of **BPmoc-F₃** initially emerged inside the island region of the agarose network, and then some of the fibers elongated from the puncta towards both the inside and outside of the agarose network and others generated and grew at void spaces of the agarose network (Supplementary Fig. 27c, Supplementary movie 5). This composite hydrogel thus has the features of both the interactive type III and the orthogonal networks. These results suggest that network patterns can be affected by the protocol of hydrogel preparation, as this reviewer points out. Furthermore, this finding highlights that the composite gel networks are not always definitely categorized into four patterns and some may be characterized as an intermediate between the four patterns. To discuss these results, we modified the main text as shown below.

Modification in the main text.

Page 20, line 14:

We also investigated whether the network patterns depend on the protocol of hydrogel preparation. A **BPmoc-F₃/Alx488-Agarose** composite hydrogel was prepared with the pH decrease protocol. CLSM imaging of this composite gel showed characteristics of both orthogonal and interactive type III networks. In this case, the agarose network maintained its original morphology, and **BPmoc-F₃** formed both well-elongated nanofibers and spherical aggregates (Supplementary Fig. 27a and 27b). Most **BPmoc-F₃** nanofibers localized independently of the agarose domains and the spherical aggregates merged with the agarose domains, which were similar to the orthogonal and interactive III networks, respectively. Time-lapse imaging of the formation process showed the agarose network formed earlier than supramolecular aggregates and fibers (Supplementary Fig. 27c, d, Supplementary Movie 5). The spherical puncta of **BPmoc-F₃** initially emerged inside the island region of the agarose network, and then some of the fibers elongated from the puncta towards both the inside and outside of the agarose network and others generated and grew at void spaces of the agarose network

(Supplementary Fig. 27c). This composite hydrogel thus has the features of both the interactive type III and the orthogonal networks. These results indicate that the network patterns may be controlled by the above-mentioned two factors, the order of network formation and interactions between supramolecular gelators/fibers and agarose polymers, while these factors can be affected by the protocol of hydrogel preparation. Furthermore, this finding highlights that the composite gel networks are not always definitely categorized into four patterns and some may be characterized as an intermediate between the four patterns.

Comment 3:

In terms of the ‘sea islands’ in the agarose gel, it can sometimes be quite difficult to see these from the imaging alone (although in other cases it is more clear). I’m not sure I would have spotted them if they hadn’t been pointed out. There is some statistical analysis in the supporting information which helped with this in distinguishing between larger and smaller void spaces.. I did wonder whether early on in the paper, it would be worth the authors providing some quantification in the paper itself of how they assessed the extent of sea-island formation in agarose alone, so that then they can benchmark this against the hybrid gels they report.

Reply 3:

According to the reviewer’s comment, we added a sentence to explain the quantification of the agarose network in the early part of the manuscript. In addition to the heterogeneity of the agarose network, we newly quantified the average sizes of the agarose island domains and void spaces (Supplementary Fig. 3). We also mentioned these quantitative analyses.

Modification in the main text.

Page 8, line 11:

To quantitatively analyze an agarose network structure, we estimated the average sizes of the island domains and void spaces by particle analyses ($0.27 \pm 0.02 \mu\text{m}^2$ and $0.51 \pm 0.03 \mu\text{m}^2$, respectively) (See Methods for detail; Supplementary Fig. 3). The homogeneity of the agarose network was also quantified by histogram analyses to evaluate the variance of the fluorescence intensity distribution (see Methods for detail; Supplementary Fig. 4).

Comment 4:

I really like the quantification of overlap and non-overlap between different gel networks that can be achieved using the CLSM technique - it is extremely powerful. The differences in behaviour between some of the supramolecular gelators is really clear. The authors also do a good job with the time-resolved studies in terms of rationalising many of their observations based on the order in which the networks form.

Reply 4:

We appreciate your positive comments.

Comment 5:

The authors contrast their observations of DBS-COOH assembly with those of Smith et al. It is perhaps worth noting that in reference 34, these researchers only noted that separate fibres of DBS-COOH and agarose could be observed by SEM – which they thus defined as ‘orthogonal’. They did not comment on whether the networks were co-localised within the wider sample, or were fully homogeneously distributed. Indeed, this would be very difficult to ascertain using their method of SEM with freeze drying. As such, Smith's report of ‘orthogonal fibres’ for this combination is not necessarily inconsistent with this paper's report that the networks co-localise. The authors could consider the wording on page 12.

Reply 5:

We do not intend to contrast Smith's and our observations. Instead, we would like to combine Smith's nanoscale SEM with our microscale CLSM to gain the deeper insight on the multiscale network structure. The combination of these imaging data indicated that the interactive type III of **DBS-COOH** and agarose would be composed of orthogonal supramolecular fibers and agarose network at a nanometer scale and the supramolecular fibers formed the micrometer-sized domain structure that was well merged with the agarose network. As this reviewer pointed out, Smith's SEM and our CLSM results are thus not necessarily inconsistent with each other. To clarify this discussion, we modified the main text as shown below.

Modification in the main text.

Page 14, line 15:

In the case of the **DBS-COOH**/agarose composite hydrogel (interactive type III), Smith's group reported the orthogonal network of **DBS-COOH** and agarose fibers with SEM images at several tens of nm resolution.³⁴ The currently conducted CLSM

imaging visualized the colocalization of the aggregated **DBS-COOH** and the island of the agarose network at sub-micrometer resolution. The combined imaging data suggested the hierarchically organized network structure of this composite hydrogel in the range from nanometer to micro/sub-millimeter scale: **colocalized micrometer-scale DBS-COOH/agarose network comprising the DBS-COOH and agarose fibers orthogonally formed at nanometer scale.**

Comment 6:

The authors report an interesting example in which they can achieve spatial resolution of one of the supramolecular gels. It is evident that over a time period of 24 hours, this system undergoes an effective phase separation, with the supramolecular gelator forming clustered aggregates. Phase separation is not unusual in polymer (gel) blends, being driven by thermodynamics of the material on the bulk scale. Some references to phase separation processes may be appropriate, and using that wording rather than ‘segregation’ may be better. It is interesting that the supramolecular gel is the one that appears to move rather than the agarose - presumably because the fibres can potentially assemble/disassemble (or are more flexible)?

Reply 6:

We appreciate your important suggestion about phase separation. As pointed out by this reviewer, time-dependent structural changes of our **NPmoc-F(F)F**/agarose composite hydrogel seem similar to phase separation processes (like an aqueous two-phase system of PEG/dextran). According to the reviewer’s suggestion, we added references about the phase separation of covalent polymers and supramolecular nanofibers and modified the main text to compare our observation with the phase separation process. In addition, we replace the words “segregate” with “phase-separate” or the related words.

To understand why the unique phase separation behavior takes place only in the **NPmoc-F(F)F**/agarose composite gel (also related to comment 7), we measured CD spectra before/after phase separation (Supplementary Fig. 30). The CD measurement revealed that the spectra became completely different before/after incubation (only the Cotton peaks of **NPmoc-F(F)F** nanofibers could be observed). It suggested that the **NPmoc-F(F)F** nanofibers before incubation are in the metastable state, which may be one of the key factors to drive phase separation.

Modifications in the main text.

Page 4, line 21:

Furthermore, we identified a unique composite hydrogel that undergoes a time-dependent network change from a homogeneously distributed pattern to a **phase-separation like** pattern at sub-millimeter scale.

Page 23, line 2:

During this study, we noticed that **only** composite hydrogel of agarose and **NPmoc-F(F)F** spontaneously changed its network at the several hundred micrometer scale during the prolonged incubation.

Page 23, line 12:

Such structural change of the **NPmoc-F(F)F/Alx488-Agarose** composite hydrogel seems similar to phase-separation processes like an aqueous two-phase system.⁵⁴⁻⁶² Notably, we found that other composite hydrogels investigated in this study did not show any time-dependent macroscopic network change for at least 48 h (Supplementary Fig. 28 and 29). To understand why the unique phase separation occurred only in the **NPmoc-F(F)F/agarose** composite gel, we investigated the packing structure of the **NPmoc-F(F)F** nanofibers before and after the phase separation by CD spectrometry (Supplementary Fig. 30). The CD measurement of the **NPmoc-F(F)F/Alx488-Agarose** hydrogel showed the Cotton peaks derived from the **NPmoc-F(F)F** nanofibers were completely different before and after the incubation (agarose did not show any Cotton peaks under this measurement condition⁴⁰) These results suggest that the as-prepared **NPmoc-F(F)F** nanofibers are in the metastable state and reassemble into a more thermodynamically stable state during aging, which may be one of the key factors to induce the phase-separation behavior.

Page 25, line 20:

To date, the similar spontaneous phase separation behaviors have been reported for multicomponent hydrogels comprising two distinct supramolecular fibers,⁶⁰⁻⁶² whereas, to the best of our knowledge, this is the first demonstration about spontaneous and fracture-induced phase separation with a combination of supramolecular fibers and covalent polymers in the field of supramolecular and polymer chemistry.

Comment 7:

Once the materials are punctured, this presumably provides a void space that will encourage the phase separation process. Phase separations always want to minimise surface energies, and the newly formed voids at the puncture points will create high

interfacial energies, which can presumably be somewhat satisfied by filling them with the supramolecular gelator – which anyway wants to phase separate – hence directing the phase separation process. I wonder whether this type of patterning should perhaps be described as 'directed phase separation', or possibly 'mechanically-directed phase separation'. I'm not sure whether such processes are well known in polymer blend materials - but it would be worth checking as there may be analogies that could be drawn. Alternatively, it may actually be something that supramolecular systems are much better set up to deliver as a result of their greater dynamics. It is, in any case, a clever way of controlling the assembled microstructure of a hybrid gel system.

Reply 7:

We really appreciate your important suggestion. According to our best efforts on literature survey, there are no reports to demonstrate mechanically-directed (fracture-induced) phase separation in the field of covalent polymer and supramolecular chemistry. Thus, this mechanically-induced phase separation is very unique. We also confirmed that phase separation did not occur in the cases of other interactive type II composite hydrogels (Supplementary Fig. 34). As described in reply 6, spontaneous structural transformation of **NPmoc-F(F)F** nanofibers from the metastable to the thermodynamic stable state may be one of the key factors to drive (mechanically-directed) phase separation.

Page 25, line 20:

To date, the similar spontaneous phase separation behaviors have been reported for multicomponent hydrogels comprising two distinct supramolecular fibers,^{60–62} whereas, to the best of our knowledge, this is the first demonstration about spontaneous and fracture-induced phase separation with a combination of supramolecular fibers and covalent polymers in the field of supramolecular and polymer chemistry.

Reviewer #2 (Remarks to the Author):

Comment 1:

In this work, Nakamura et al. investigates different supramolecular hydrogel composites made of agarose and various low molecular weight (LMW) gelators. Using in-situ confocal imaging, the different localization of agarose and LMW gelators were characterized and categorized into four different network patterns. These network patterns were claimed to be governed by the order of network formation and interaction between supramolecular gelators/fibers and agarose. Although the phenomena observed between the different systems is interesting, the current work lacks the discussion to

rationalize the observation presented by the authors. Additional studies also need to be done to support the claims of the authors in different parts of the paper. The authors also do not make an attempt to relate the structures of their materials to any sort of physical attributes, which is important for the context and rationale behind this work. This paper is therefore not recommended for publication in Nature Communications. The following comments are listed below to help the authors improve the paper.

Reply 1:

We appreciate your careful reviewing of our manuscript. According to your comments, we conducted additional experiments about rheological analyses of composite hydrogels, time-lapse CLSM imaging of formation processes of single-component hydrogels, and fracture-induced experiments with other composite hydrogels, *etc.* We carefully amended the main text and supplementary information to rationalize our observations and claims. The detailed results were explained in point-by-point response as shown below. We believe that our modified manuscript is suitable for publication in Nature Communications. All modifications in the main text and supplementary information are highlighted in red letters.

Comment 2:

While the CLSM images show a difference in morphology of the separate network patterns, there is no difference observed between the SEM images. Therefore, the entirety of the work is relying on the given set of CLSM imaging. It would be beneficial to have at least one more method to observe morphological differences between the samples. A possible solution could be to use Cryo-FIB SEM. This can better preserve the morphology of the structure and eliminates drying effects that can be generated when using lyophilization.

Reply 2:

We would like to emphasize that the findings presented in this work could not be obtained with SEM imaging. SEM imaging cannot discriminate between supramolecular and agarose fibers due to the morphological similarity of their shapes and diameters. Furthermore, selective staining methods for SEM imaging have not been developed. In sharp contrast, CLSM imaging can distinguish supramolecular nanofibers from agarose network by use of selective fluorescent probes and fluorescently modified agarose. We thus believe that our CLSM data would be sufficient to categorize the network patterns of the composite hydrogels.

Comment 3:

In the introduction, the authors highlight the importance of different network patterns on the mechanical performance of tissues (tension, compression). Did the authors try to investigate whether there were any mechanical differences for each of the network patterns they claimed to achieve? Are there any rheological investigations performed to see how the networks differ in their viscoelastic properties? Tensile, compressive and rheological testing for each of the different networks would improve the impact of this work.

Reply 3:

To address the reviewer's concern, we measured the viscoelastic properties of the four representative composite hydrogels with distinct network patterns (**BPmoc-F₃**/Agarose, **Phos-MecycC₅**/Agarose, **GalNAc-cycC₆**/Agarose, and **DBS-COOH**/Agarose) and the corresponding single-component hydrogels (Agarose, **BPmoc-F₃**, **Phos-MecycC₅**, **GalNAc-cycC₆**, and **DBS-COOH**) presented in Figure 2 (Supplementary Fig. 15 and 16). In linear dynamic mechanical tests, the storage modulus (G') for all the composite gels was appreciably higher than the loss modulus (G''), and both G' and G'' were almost constant in a frequency range between 0.1 and 10 rad s⁻¹, confirming they are "hydrogels" without flowability (Supplementary Fig. 15). The tan δ values were almost comparable among the composite hydrogels. We noticed that **GalNAc-cycC₆**/Agarose showed a higher G' value (6448 Pa) than the other composite hydrogel (1240–1749 Pa). To quantitatively assess the effect of mixing different gels on stiffness, we defined an enhancement factor as the ratio of the G' value for the composite hydrogel over the sum of G' values for single component gels. To our surprise, we found that the enhancement factor of **GalNAc-cycC₆**/Agarose was 5.4 while those of the other composite gels were below 2.5, suggesting the stiffness of the interactive type I was synergically increased (Supplementary Fig. 15l). The tan δ values of the composite hydrogels were slightly higher than that of the single-component agarose hydrogel probably due to liquid-like property of supramolecular networks. We also investigate the nonlinear viscoelastic properties of the composite gels to determine the yield strain, defined as the G'/G'' cross-over point, in the amplitude sweep (Supplementary Fig. 16). The yield strain of **GalNAc-cycC₆**/Agarose (11.6%) was lower than those of the other composite gels (above 42%), indicating that the **GalNAc-cycC₆**/Agarose gel (interactive type I) was stiffest and most brittle among the composite hydrogels we tested (Supplementary Fig. 16k and l). These results suggest that the viscoelastic property may depend on the network patterns. To

discuss these viscoelastic properties, we modified the main text and supplementary information as shown below.

The main claim of this paper is the finding of the four distinct network patterns of supramolecular-polymer composite hydrogels, which are controlled by the order of formation and interactions. Therefore, we believe that the detailed rheological studies including other tensile and compressive tests are out-of-scope of this paper. We will carefully investigate the detailed relationship between the network patterns and rheological properties in the future.

Modifications in the main text.

Page 1, line 4, 8:

We added new authors, Takuma Aoyama and Kenji Urayama, due to their contribution of rheological experiments and analysis.

Page 16, line 2:

Rheological properties of composite hydrogels with four distinct network patterns

We investigated the viscoelastic properties of these four representative composite hydrogels with distinct network patterns. In linear dynamic mechanical tests, the storage modulus (G') for all the composite gels was appreciably higher than the loss modulus (G''), and both G' and G'' were almost constant in a frequency range between 0.1 and 10 rad s^{-1} , confirming they are elastic hydrogels without flowability (Supplementary Fig. 15). We noticed that **GalNAc-cycC₆/Agarose** showed a higher G' value (6448 Pa) than the other composite hydrogels (1240–1749 Pa) (Supplementary Fig. 15k). To quantitatively assess the effect of mixing different gels on stiffness, we defined an enhancement factor as the ratio of the G' value for the composite hydrogel over the sum of G' values for single component gels. To our surprise, the enhancement factor of **GalNAc-cycC₆/Agarose** was 5.4 while those of the other composite gels were below 2.5, suggesting the stiffness of the **GalNAc-cycC₆/Agarose** gel is synergically increased (Supplementary Fig. 15l). The nonlinear viscoelastic properties of the composite gels were also examined to determine the yield strain, defined as the G''/G' cross-over point, in the amplitude sweep experiment (Supplementary Fig. 16). The yield strain of **GalNAc-cycC₆/Agarose** (11.6%) was lower than those of any other composite gels (42.0–82.1%). Taken together, the **GalNAc-cycC₆/Agarose** gel is stiffest and most brittle among the composite hydrogels we tested, suggesting the network patterns of the composite hydrogels might give impacts on their rheological properties.

Comment 4:

On page 5, when the different LMW gelators were introduced, there were no explanation as to why the specific molecular structures were selected for the study. Discussing it at this point of the paper will help the reader.

Reply 4:

Thank you for your suggestion. All LMW hydrogelators used in this manuscript (except for **DBS-COOH**) were picked up from our previous reports. The peptide-type hydrogelators possess different stimuli-responsive *N*-terminal moieties and the optimal critical gelation concentrations. Also, we chose the lipid-type hydrogelators with chemically-distinct hydrophilic head groups because the head groups locate at the surface of supramolecular fibers to interact with the agarose network. It is reasonably expected that the hydrophobic tail groups would show negligible impacts on interaction with agarose because they are buried inside the nanofibers. We added the sentence to clarify why the specific molecular structures were selected.

Modifications in the main text.

Page 5, line 8

Three peptide-type LMW gelators with a di- or triphenylalanine sequence and a distinct *N*-terminal moiety [boronophenyl (BPMoc)²², nitrophenyl (NPMoc)²⁶, and benzaldehyde (Ald) derivatives⁴³] and three lipid-type gelators with the different head groups [phosphate (Phos)⁴³, GalNAc⁴⁴, and Lys⁴⁵] were picked up from our small library of LMW gelators, as shown in Figure 1b and Supplementary Fig. 1.

Page 5, line 17:

According to our previous report,⁴⁸ the head groups locate at the surface of supramolecular fibers and are exposed to the water phase. We thus examined the effects of the head groups rather than the hydrophobic tail groups, which are buried inside the nanofibers, on the composite network structures through the interaction with the agarose network.

Comment 5:

There are only single line plots presented for each overlay of the CLSM image. Additional line plots should be provided in the supporting information.

Reply 5:

According to the reviewer's suggestion, additional line plots were provided in the representative CLSM images (Supplementary Fig. 5–8). We confirmed that the tendency in peak patterns of these plots is identical to those shown in the main text. To quantitatively analyze the peak patterns, we further calculated the peak distance between the adjacent supramolecular nanofiber and agarose domain (Supplementary Fig. 9). These quantitative analyses showed that the network distances in interactive type I and type III was 120 ± 110 nm and 150 ± 170 nm, respectively, which were significantly lower than the orthogonal and interactive type II (200 ± 200 nm and 210 ± 170 nm, respectively). These results support our claim that supramolecular and agarose networks are localized closely with each other in the interactive type I and type III.

We added these results and discussion to the main text and supporting information.

Modifications in the main text.

Page 9, line 9

The average nearest peak distance between the supramolecular and agarose network was estimated to be 120 ± 110 nm, which was statistically smaller than that of the orthogonal **BPmoc-F₃/Alx488-Agarose** (200 ± 200 nm) (Supplementary Fig. 5, 6, 9).

Page 10, line 2:

The average low Pearson's correlation coefficient (0.01 ± 0.02) and line plot analysis (peak distance: 210 ± 170 nm) indicated that the two networks were not correlated with each other (Supplementary Fig. 7 and 9).

Page 10, line 19:

The average PCC value (0.23 ± 0.01) and line plot analyses (peak top distance: 150 ± 170 nm) indicate that the core of the **DBS-COOH** aggregates were overlapped with the agarose network, and the thinner **DBS-COOH** fibers at the periphery of the aggregates were in the void space of the agarose (Supplementary Fig. 8 and 9).

Comment 6:

In general, additional CLSM micrographs in the supporting information will help the reader and reviewers to determine if the presented micrographs in the main text are representative of the sample as a whole.

Reply 6:

We added two more CLSM images of each composite hydrogel in the supplementary information (Supplementary Fig. 12–14). The network structures in these images are similar to those in the main text. Thus, the CLSM images shown in the main text are representative of the sample as a whole.

Modification in the main text.

Page 13, line 6:

We obtained two more CLSM images of each composite hydrogel and confirmed that the network structures in these images are similar to those shown in Figure 2. Thus, the CLSM images shown here are the representative ones. The additional images are shown in Supplementary Fig. 12–14.

Comment 7:

The schematic illustrations of the interactive type I are slightly misleading, as it does not seem clear from the CLSM image that the agarose aligns with the LMW gelator fibers as is drawn in the schematic. Additionally, it is hard to see a clear difference in pattern between orthogonal and interactive type II from the CLSM images, as opposed to what the schematic illustrations suggest.

Reply 7:

To avoid such confusion, we modified the illustrations to describe the interactive type I and type II networks more properly.

Comment 8:

A pH dependence experiment for the DBS-COOH composite is missing.

Reply 8:

The hydrogel preparation by reducing the pH completely followed the Smith's protocol. We newly measured time-course of pH change during the hydrolysis of glucono- δ -lactone (hydrogelation) (Supplementary Fig. 10). pH value gradually decreased from 7.3 to 5.2 in 3 h. Combined with real-time CLSM imaging data, the nanofiber formation was initiated at pH 6.7. The pK_a value of **DBS-COOH** was determined to be 5.4 in the previous report [Cornwell, D. J. *et al. J. Am. Chem. Soc.* **137**, 15486 (2015)]. These results suggest that the *ca.* 5% protonation of the **DBS-COOH** may reach the critical aggregation concentration to initiate the nanofiber formation.

Modification in the main text.

Page 10, line 8

According to Smith's protocol, the composite hydrogel of **DBS-COOH/Alx488-Agarose** containing **TMR-Gua** was prepared by reducing the pH via hydrolysis of glucono- δ -lactone (termed a pH decrease protocol, **the time course of pH change was shown in Supplementary Fig. 10**).

Comment 9:

SEM micrographs show images of the gel after lyophilization whereas CLSM are obtained in its hydrated state. This must be clearly distinguished in the paper.

Reply 9:

According to the reviewer's comment, we modified the main text and caption in Figure 2 and 3 to clarify the state of the samples.

Modifications in the main text.

Page 13, line 9:

All of CLSM images of hydrogels were acquired in the hydrated state.

Page 14, line 3:

SEM imaging was used to further evaluate the structures of the composite hydrogels. Single-component gels of agarose, **BPmoc-F₃**, **Phos-MecycC₅**, and **GalNAc-cycC₆** were lyophilized **and analyzed in the dried state** with SEM, revealing that nanofibers with a diameter of 20–150 nm were bundled to construct the network structures for all the four cases (Figure 3a). In the **freeze-dried BPmoc-F₃/agarose** composite hydrogel, the SEM images showed many fibers with a diameter of 20–100 nm (Figure 3b, left). Because there were little differences in the morphology, it was difficult to distinguish between the supramolecular fibers and agarose fibers. In the **freeze-dried Phos-MecycC₅/agarose** and **freeze-dried GalNAc-cycC₆/agarose** gels, indistinguishable nanofibrous structures, such as the **BPmoc-F₃/agarose** gel, were also observed (Figure 3c and 3d).

Page 15, line 9:

The freeze-dried samples were used for SEM imaging.

Comment 10:

Were kinetics studies of the single component hydrogels performed? The only ones presented were that of the supramolecular composites.

Reply 10:

According to the reviewer's suggestion, we newly conducted time-lapse CLSM imaging of formation process of the single component hydrogels (Supplementary Fig. 21). We added results and discussion about the kinetics study of the single-component hydrogels. Please also refer to Reply 12 for the detailed results.

Modification in the main text.

Page 19, line 24

We also confirmed no significant differences in the formation kinetics of supramolecular fibers and agarose network between the single-component and composite hydrogels in all cases, implying that the interaction between supramolecular gelators/fibers and agarose is not strong enough to alter the formation kinetics (Supplementary Fig. 21).

Comment 11:

The authors claim that the formation processes for the composite hydrogels of agarose and LMW gelators is governed by the interactions between the gelator and the agarose. However, there are no discussion on how the different chemical structure of the LMW gelator differed in their interaction with the agarose. A detailed rationale of the interaction between the gelators and fibers, and how they related to the resulting structure of the composite is needed.

Reply 11:

In the case of orthogonal and interactive type III (agarose network forms first), hydrophobicity of supramolecular hydrogelators may be important. In **BPmoc-F₃**/agarose (orthogonal), **BPmoc-F₃** has a negative charge due to deprotonation of carboxylic acid at neutral pH, resulting in decreasing hydrophobic interaction with the agarose network. On the other hand, in **DBS-COOH**/agarose (interactive type III), carboxylate groups of **DBS-COOH** are protonated at acidic condition induced by hydrolysis of glucono- δ -lactone, which may enhance interaction between the agarose network probably through hydrophobic interaction and/or hydrogen bonding. Thus, protonated **DBS-COOH** can nucleate near/inside the agarose network to form the interactive type III network.

In the case of interactive type I and II (supramolecular fibers form first), the surface structure of supramolecular fibers may determine the network structure. In the case of

GalNAC-cycC₆/agarose (interactive type I), **GalNAC-cycC₆** fibers present lots of sugar moiety on their surface that can interact with agarose polymers. Such polyvalent sugar-sugar interaction may be one of the keys for formation of interactive type I. In **Phos-cycC₆/agarose** (interactive type II), the negative **Phos-cycC₆** hydrogelator forms polyanionic supramolecular nanofibers, which may cause a negligible interaction with agarose polymers. Despite negligible chemical interaction between **Phos-cycC₆** and agarose fibers, **Phos-cycC₆** fibers may physically suppress diffusion of agarose polymers, resulting in more homogeneous agarose network (interactive type II).

To discuss relationship between chemical structure and interaction, we modified the main text as shown below.

Modification in the main text.

Page 18, line 25:

For **BPmoc-F₃/Alx488-Agarose** and **DBS-COOH/Alx488-Agarose** hydrogels, hydrophobicity of the supramolecular hydrogelators may be important for the interactions. In **BPmoc-F₃/Alx488-Agarose** (orthogonal), **BPmoc-F₃** has a negative charge due to deprotonation of carboxylic acid at neutral pH, resulting in decreasing hydrophobic interaction with the agarose network. On the other hand, in **DBS-COOH/Alx488-Agarose** (interactive type III), carboxylate groups of **DBS-COOH** are protonated at acidic conditions induced by hydrolysis of glucono- δ -lactone, which may enhance interaction between the agarose network probably through hydrophobic interaction and/or hydrogen bonding. Thus, protonated **DBS-COOH** can nucleate near/inside the agarose network to form the interactive type III network.

Page 19, line 17:

For **GalNAC-cycC₆/Alx488-Agarose** and **Phos-MecycC₅/Alx488-Agarose**, the surface structure of supramolecular fibers may determine the network structure. In **GalNAC-cycC₆/Alx488-Agarose** (interactive type I), **GalNAC-cycC₆** fibers present lots of sugar moiety on their surface that can interact with agarose polymers through polyvalent sugar-sugar interaction, resulting in interactive type I. In **Phos-MecycC₅/Alx488-Agarose** (interactive type II), **Phos-MecycC₅** forms polyanionic supramolecular nanofibers due to a negative phosphate group, which may cause a negligible interaction with agarose polymers. We also confirmed no significant differences in the formation kinetics of supramolecular fibers and agarose network between the single-component and composite hydrogels in all cases, implying that the interaction between supramolecular

gelators/fibers and agarose is not strong enough to alter the formation kinetics (Supplementary Fig. 21).

Comment 12:

In addition, the authors also differentiated between the (i) order of network formation and (ii) interaction between the gelators and agarose as the basis for network formation of the composite. However, would the network of formation not be heavily influenced by the interaction between the gelators and agarose as well? The kinetics study of the single component hydrogel would provide good supporting information with this part of the study.

Reply 12:

Thank you for your important comment. As described in reply 10, we conducted the kinetics study of the single-component hydrogel (Supplementary Fig. 21). There are no significant differences in the formation kinetics between the single-component and composite systems. These data suggested that interaction between supramolecular gelator/fibers and agarose is not strong enough to alter the formation kinetics.

We added results and discussion about the kinetics study of the single-component hydrogels.

Comment 13:

Would the authors expect the same fracture-induced remodelling for other supramolecular hydrogel composites tested other than the NPmoc-F(F)F/Alx488-Agarose?

Reply 13:

We newly performed CLSM imaging of all other interactive type II composite hydrogels at 0 and 16 h after fracture with the needles (Supplementary Fig. 34). The remodeling did not occur in the other interactive type II composite hydrogels (**Phos-MecycC₅/Alx488-Agarose**, **Lys-cycC₅/Alx488-Agarose**, **0.4 wt% BPmoc-F₃/Alx488-Agarose**) except **NPmoc-F(F)F/Alx488-Agarose**. Thus, this dynamic remodeling behavior is unique to **NPmoc-F(F)F/Alx488-Agarose**.

We added these results and discussion in the main text and supplementary information.

Modification in the main text.

Page 24, line 19:

Notably, such network remodeling was not observed in the 24 h aged **NPmoc-F(F)F/Alx488-Agarose** composite hydrogel **and the other composite hydrogels with the interactive network type II** (Supplementary Fig. 33 and 34).

Comment 14:

Page 5, lines 13-16 needs citation and supporting literature.

Reply 14:

We added two references in the paragraph to support this statement.

Modification in the main text.

Page 5, line 12:

The peptide-type gelators self-assemble into β -sheet like nanofibrous structures via hydrogen bonding and π - π interactions between the phenylalanine peptides.⁴⁶ The lipid-type gelators are amphiphiles comprising a hydrophilic head and hydrophobic cycloalkane tail group that form one-dimensional nanofibers mainly via hydrophobic interactions in an aqueous solution.⁴⁷

Reviewer #3 (Remarks to the Author):

Comment 1:

The authors provide important new insight into the assembly of hydrogel networks between a polymeric hydrogel and many small molecule hydrogels. Confocal light scanning microscopy is the main tool used for their analysis. The authors distinguish several different morphologies which they ascribe to the kinetics of gel formation and interactivity between the gels. They also explore the dynamic reorganization of one of the gel systems in good detail. The work provides a methodology for the identification of several different hydrogel network types using a simple CLSM setup. Furthermore, they give rationale for how each of the hydrogel networks can be targeted, through the adjustment of the kinetics of the assembly of the hydrogels, as well as the interactivity between the gels. This is a significant advancement in hydrogel network design. The experiments appear to be conducted properly. Experimental detail is sufficient for reproduction, though the number of trials/images used for data collection should be specified.

The goal of the research project is good and the authors have designed some effective

systems to achieve their goal of hydrogel network classification and understanding of the mechanisms by which different networks are formed. However, the evidence sometimes does not fully support the claims and, in some cases, even contradicts their conclusions. At the crux of my concerns is their classification of the various networks. The rationale why they chose to classify some networks as orthogonal, interactive, and the different types of interactive networks is not clearly explained, nor is it entirely clear as to the exact cutoffs for the classification of each type of network. With these concerns, the manuscript needs major revision before acceptable for publication. Some specific comments are summarized below:

Reply 1:

We appreciate your careful review and comments on our manuscript. To address the reviewer's concerns about the classification, we conducted additional quantitative image analyses, including the sizes of the agarose domain/void, and the distance between individual networks. By combination of PCC values, domain/void sizes and homogeneity of agarose, and peak distance between supramolecular fibers and agarose, we provide a potential classification guide for the composite networks (see reply to comment 16 in detail). We modified the main text to explain our quantitative image analyses. We believe that such the detailed image analysis offers the rationale for our network quantification and our modified manuscript is suitable for publication in Nature Communications. All modifications in the main text and supplementary information are highlighted in red letters.

Comment 2:

For BPmoc-F3 and Agarose (Figure 2b) the Pearson's coefficient is 0.1, which indicates weak to no correlation between the two assemblies. However, this is not exactly what is shown in the schematic diagram used to represent this system (Figure 2c). The diagram shows an "Orthogonal" distribution of the two assemblies, which would correspond to a negative Pearson's correlation coefficient. What is the reasoning for this?

Reply 2:

In our Airyscan CLSM images, there seems a numerous number of intersections of supramolecular fibers and agarose, even if orthogonal network, because of the limited z-resolution of 350 nm. Such overlap may cause the PCC values to be nearly zero. In order to determine **BPmoc-F₃**/agarose network as the orthogonal network, further image analyses (not only PCC values but also line plot analysis and histogram analysis) and real-

time observation of formation process are essential (please also refer to Reply 5 for Reviewer 2).

To make our manuscript more precise, we replace the word “segregated” into “no correlation” or the related phrases.

Modifications in the main text.

Page 8, line 21:

The overlay image showed that the two networks were **not correlated with each other**;

Comment 3:

Figure 2e has a faint “G” in the center of the schematic. Either explain its inclusion or remove it if it is an error.

Reply 3:

Thank you for paying careful attention. This is just an error, and we removed it.

Comment 4:

While there is obvious interactive network formation between GalNAc-cycC6 and Agarose (Figure 2d), the PCC remains quite low for what is shown as a near one-for-one assembly in Figure 2e. Why is the PCC so low in this situation?

Reply 4:

This is probably because some of the agarose domains were well merged with the supramolecular network but others did not overlap, resulting in the PCC values of about 0.3.

Comment 5:

Figure 2f shows a PCC of 0.02, but the text (Page 8, Line 19) indicates a PCC of -0.01. Please correct this discrepancy.

Reply 5:

0.02 is the correct value. We corrected the main text.

Comment 6:

5) Page 8, Line 19 “The low Pearson’s correlation coefficient (-0.01) and line plot analysis indicated that the two networks were well segregated.” A strongly negative PCC would indicate segregation, but a PCC near 0 indicates no correlation.

Reply 6:

As mentioned in the reply 2, we modify the word, “segregate,” to a more appropriate one, such as no correlation.

Comment 7:

6) The PCC of Interactive Type II network is smaller (-0.01) than that of orthogonal network (0.10). Since this seems to be a large part of the basis for your classifications, why do you ascribe orthogonal characteristic to one and interactive characteristic to the other?

Reply 7:

We categorize the network patterns into orthogonal or interactive by comparing supramolecular/agarose networks of the composite gel with those of single-component gels. If the networks of the composite gel are identical with those of the single-component gels, the network is classified into the orthogonal network; otherwise, the network is classified into the interactive network. To quantitatively classify the network patterns, we further analyzed homogeneity and island/void sizes of the agarose network (table R1, see reply 16 for the detailed discussion about the classification guide). In the case of the interactive type II network, homogeneity and island/void sizes (as described in reply 8) become smaller than the single-component agarose gel. We thus classified **Phos-MecycC₅**/agarose gel into the interactive network despite no correlation between two networks, as confirmed by the PCC value.

Table R1. A guideline for classification of supramolecular/polymer composite hydrogels

	PCC	Homogeneity ^a	Island size (μm ²) ^a	Void size (μm ²) ^a
Agarose	–	(1.373 ± 0.010)×10 ⁴	0.27 ± 0.02	0.51 ± 0.03
Orthogonal	<0.2	Not significant	Not significant	Not significant
Interactive I	>0.2	Smaller	changed into fibrous morphology	
Interactive II	<0.2	Smaller	Smaller	Smaller
Interactive III	>0.2	Not significant	Not significant	Not significant

a: statistical difference against the single-component agarose

Modification in the main text.

Page 10, line 4:

Since the agarose morphology was altered from that of the single-component gel, this network is referred to as interactive network type II (Figure 2g).

Comment 8:

7) For Interactive Type II (Figure 2f,g) the claim is made that the void size in the agarose sea-island network decreased so that there is a more uniform distribution when compared to single component agarose, such that the agarose islands are the same size but the distance between islands is smaller. However, the schematics 2c and 2g suggest that the agarose islands are decreasing in size along with the space between the islands getting smaller as well. Could you clarify if the claim is that both the island and void space is getting smaller, or just the void space, and adjust the figures accordingly? If the islands are getting smaller, particle size quantification would be helpful in backing up this claim.

Reply 8:

Thank you for your important suggestion. We conducted the particle analysis of both the islands and the void spaces of the agarose network (Supplementary Fig. 3). The void sizes in the interactive type I (**GalNAc-cycC₆**/Agarose) and type II (**Phos-MecycC₅**/Agarose) networks (0.358 ± 0.017 and $0.32 \pm 0.03 \mu\text{m}^2$, respectively) were significantly smaller than those in the single-component agarose ($0.51 \pm 0.03 \mu\text{m}^2$), orthogonal (**BPmoc-F₃**/Agarose, $0.50 \pm 0.06 \mu\text{m}^2$), and type III networks (**DBS-COOH**/Agarose, $0.50 \pm 0.12 \mu\text{m}^2$). These results are consistent with our description of the change in the agarose network, as confirmed by histogram analysis. We also found that the average island size was decreased in Interactive type II ($0.210 \pm 0.006 \mu\text{m}^2$) when comparing with the single-component agarose gel ($0.27 \pm 0.02 \mu\text{m}^2$). On the other hand, there are no significant differences in the average island sizes of the orthogonal (**BPmoc-F₃**/Agarose, $0.244 \pm 0.003 \mu\text{m}^2$) and interactive type III (**DBS-COOH**/Agarose, $0.245 \pm 0.003 \mu\text{m}^2$) against the single-component agarose gel. These observations can be applicable to other composite hydrogels. Therefore, we concluded that these results showed that both the void space and island structures get smaller in interactive type II.

Modifications in the main text.

Page 9, line 24:

The particle analysis quantitatively revealed that the average size of the agarose island domains ($0.211 \pm 0.006 \mu\text{m}^2$) was smaller than the single component agarose ($0.27 \pm 0.02 \mu\text{m}^2$) or the orthogonal **BPmoc-F₃/Alx488-Agarose** ($0.244 \pm 0.003 \mu\text{m}^2$) (Supplementary Fig. 3b).

Page 10, line 16:

On the other hand, the agarose showed the sea–island network, nearly identical to those of the single component as confirmed by the histogram analysis [$s = (1.37 \pm 0.03) \times 10^4$] and the particle analysis (domain size: $0.245 \pm 0.003 \mu\text{m}^2$, pore size $0.51 \pm 0.03 \mu\text{m}^2$) (Supplementary Fig. 3 and 4).

Comment 9:

8) Page 9, Line 9: “This behavior is different from the above-mentioned network patterns, where the morphology of the supramolecular fibers network significantly changed with negligible change of the agarose network.” I believe that this sentence is improperly worded, as it gives the impression that the previous networks show no change in agarose assembly.

Reply 9:

According to your suggestion, we modified the sentence as shown below.

Modification in the main text:

Page 10, line 23:

This behavior is different from the above-mentioned network patterns, and the morphology of the supramolecular fibers network significantly changed with negligible change of the agarose network.

Comment 10:

The scale of the images used in the analysis is $\sim 25 \mu\text{m} \times 25 \mu\text{m}$. Since the image analysis is the crux of the manuscript, it is important for the reader to know how many images/how large of an area was surveyed in each case. Please indicate how many images were used in each analysis.

Reply 10:

We repeated each experiment at least 3 times and confirmed the reproducibility of each image. Regarding the network pattern classification, we newly used the average value of 3 images.

We modified the main text, Figure, and supplementary information to clarify how many images are used in each analysis or experiment.

Modification in the main text.

Page 13, line 6:

We obtained two more CLSM images of each composite hydrogel and confirmed that the network structures in these images are similar to those shown in Figure 2. Thus, the CLSM images shown here are the representative ones. The additional images are shown in Supplementary Fig. 12–14.

Comment 11:

Page 15, Line 17: “The agarose network may assist the formation of the DBS-COOH fibers, likely through interactions between DBS-COOH and agarose. Therefore, interactions between two fibers/networks may be an additional controlling factor.” A control experiment comparing the kinetics of DBS-COOH assembly with the co-assembly would be helpful in assessing the validity of this claim. For the interactive type networks, there should be some rate differences between the co-assembly and lone assemblies of the gelators/agarose.

Reply 11:

Thank you for your important comment. We newly conducted real-time CLSM imaging of formation processes of the single-component hydrogel (Supplementary Fig. 21). There are no significant differences in the formation kinetics between the single-component and composite systems. These data suggested that the interaction between supramolecular gelators/fibers and agarose is not strong enough to alter the formation kinetics.

We added results and discussion about the kinetics study of the single-component hydrogels.

Modification in the main text.

Page 19, line 24

We also confirmed no significant differences in the formation kinetics of supramolecular fibers and agarose network between the single-component and composite hydrogels in all

cases, implying that the interaction between supramolecular gelators/fibers and agarose is not strong enough to alter the formation kinetics (Supplementary Fig. 21).

Comment 12:

If Interactive Network Type II (Figure 4i, Page 16, Line 6) is the result of non-interaction between the agarose and LMW gelator, why it is called interactive?

Reply 12:

As described in reply 7, we define the composite gel network with the distinct morphology from the single-component gels as an interactive one. As for interactive type II, we expected that there are negligible chemical interactions between supramolecular fibers and agarose but supramolecular fibers would physically suppress diffusion of agarose fibers to decrease the mesh size of the agarose network, as mentioned in the main text.

Comment 13:

The experiment in which BPmoc-F3 concentration is increased to 0.4 wt% shows an obvious and significant change in supramolecular assembly. The authors claim that this is a change from Orthogonal to Interactive Type II network pattern. However, the PCC of the new network is becoming more negative (-0.07) indicating a network that is more orthogonal. This further throws into question the naming convention of the proposed networks, why is this negatively correlated network named Interactive? Is there a further increase in the negative correlation upon further increase of the BPmoc-F3 concentration?

Reply 13:

As described in reply 7 and 12, we define the composite gel network with the distinct morphology from the single-component gels as an interactive one. To quantitatively compare PCC values between orthogonal and interactive type II, we acquired more CLSM images (Supplementary Fig. 12–14). As a result, there was no significant difference in the average PCC values between the 0.1 wt% **BPmoc-F₃**/Agarose (0.08 ± 0.10) and 0.4 wt% **BPmoc-F₃**/Agarose (0.04 ± 0.13). We also tried to make a composite hydrogel containing more than 0.4 wt% **BPmoc-F₃**, but it was very difficult to handle due to too rapid formation of the hydrogel. To explain these experimental results, we modified the main text and supplementary information as shown below.

Modification in the main text.

Page 20, line 5:

For example, the network pattern of **BPmoc-F₃/Alx488-Agarose** changed from the orthogonal to the interactive type II when the concentration of **BPmoc-F₃** was increased from 0.1 wt% to 0.4 wt% (PCC values: 0.08 ± 0.10 and 0.04 ± 0.13 , respectively; Supplementary Fig. 25).

Comment 14:

Page 16, Line 19: “Time-lapse CLSM imaging of the composite hydrogel containing 0.4 wt% BPmoc-F₃ revealed the order of the network formation was reversed; specifically, the nanofiber formation of BPmoc-F₃ was faster than the agarose network formation (Supplementary Fig. 13).” The figure in questions shows that both agarose and supramolecular fibers exist at the start of the assembly process. Based on the frames shown, it appears that the agarose gel is fully formed at 54 s, while the supramolecular fiber is only beginning to form at this point. This appears to go directly against the claims in the body of the manuscript. Is there an explanation for this?

Reply 14:

At 54 s, the agarose channel image showed homogeneously-distributed fluorescence, suggesting that agarose did not form its network at this time point, albeit high fluorescence intensity. In contrast, the TMR-Gua channel clearly visualized the nanofibrous structures of **BPmoc-F₃**. Thus, we conclude that the nanofiber formation of **BPmoc-F₃** was faster than agarose network formation. Possible reasons why the agarose channel showed higher fluorescence intensity are unintended drift of the focus plane and/or changes in refractive index due to high sample temperature.

To minimize effects of fluctuation of the fluorescence intensity, we re-analyzed the time course change of agarose network formation using the particle analysis to count the agarose domains (Supplementary Fig. 26). This analysis showed that the number of the agarose domains increased during 600 min, which supported our proposed mechanism; supramolecular fibers formed faster than agarose network in interactive type II. This new analysis method is able to be applied to other composite hydrogels. Therefore, we believe that these results are well consistent with our proposed mechanism.

Modifications in the main text.

Page 17, line 8:

The **number** of the agarose domains gradually increased and reached a constant within 5 min (Figure 4b, green).

Comment 15:

The gel remodeling experiments are very interesting, elegant, and thoroughly explored. However, they raise an issue that I have mentioned a few times in my previous comments. Specifically, it appears that Interactive Type II networks are in fact repulsive/exclusionary in nature, but are kinetically trapped in some metastable state that appears slightly orthogonal on the small scale. Upon the fracturing, the gels reassemble into their preferred exclusionary assembly, as evidenced by the many micro-needle experiments and the strongly negative PCC (-0.42). Based on Supplementary Figure 14, the PCC of the NPmoc-F(F)F system is -0.52 when measured at the same scale as the other systems. This suggest to me that this is not an interactive system driven by favourable interactions, but rather an exclusionary assembly driven by repulsion. Is the reorganization behavior common to all Interactive Type II systems or only the NPmoc-F(F)F hydrogelator? Do you see reorganization for the orthogonal networks under similar conditions?

Reply 15:

As pointed out by this and the other reviewers, time-dependent structural changes of our NPmoc-F(F)F/Agarose composite hydrogel seem similar to phase separation (like an aqueous two-phase system of PEG/dextran). We added references about the phase separation of covalent polymers and supramolecular nanofibers and modified the main text to compare our observation and the phase separation process.

To answer the reviewer's questions, we obtained the CLSM images of all other interactive type II composite hydrogels at 0 and 16 h after fracture with the needles (Supplementary Fig. 34). As results, the remodeling did not occur in the other composite hydrogels except NPmoc-F(F)F/Alx488-Agarose. The orthogonal network did not show spontaneous network remodeling during aging (Supplementary Fig. 29). Therefore, we concluded that this dynamic remodeling behavior is unique to the NPmoc-F(F)F/Alx488-Agarose composite hydrogel.

Modification in the main text.

Page 24, line 19:

Notably, such network remodeling was not observed in the 24 h aged NPmoc-F(F)F/Alx488-Agarose composite hydrogel and the other composite hydrogels with the interactive network type II (Supplementary Fig. 33 and 34).

Comment 16:

The classification of the networks, while ambitious, appears to have several issues which keep it from being a robust classification protocol. The exact method by which the classifications were arrived at is quite vague in the text and remains largely qualitative. If the goal of the manuscript is to provide the reader with a universal method for classifying various hydrogel networks, a more rigorous, step-by-step guide should be provided. Perhaps better classification would be to use Orthogonal, Interactive Type I, Interactive Type II, and Exclusionary (Or Repulsive) to account for the negative PCC observed in some of the networks.

Reply 16:

We would like to emphasize that our goal is not to provide the reader with a universal method for classifying various hydrogel networks. Our main claim is the discovery of distinct network patterns in the supramolecular/polymer composite hydrogels.

As pointed out by this reviewer, however, the classification guide would be useful for readers. We thus would like to discuss the potential classification guide. According to our quantitative image analyses, the network patterns can be classified based mainly on Pearson's correlation coefficient values, the standard deviation value and the island/void sizes of the agarose network (table R1). We can set the criteria of PCC value at 0.2 to differentiate the network patterns: If the value of Pearson's correlation coefficient is lower than 0.2, the composite hydrogels were classified into orthogonal or interactive type II; otherwise, interactive type I or III. The network patterns are further divided by use of the standard deviation value and the island/void sizes of the agarose network. Compared to the single-component agarose gel, the network can be classified into the interactive type I or II if those values are significantly smaller; otherwise, orthogonal or interactive type III (in the interactive type I, the agarose network obviously changes into the fibrous structure).

However, it has not been confirmed whether this classification can be applicable as a universal guide because of some exceptional composite hydrogels and the small sample size. For example, the **BPmoc-F₃**/Agarose hydrogel prepared by the pH decreasing protocol shows an intermediate network that exhibits characteristics of both orthogonal and interactive III networks (see reply 2 in reviewer 1 in detail). This finding highlights that the composite gel networks are not always definitely categorized into four patterns and some may be characterized as an intermediate between the four patterns. Therefore, the network classification should be carefully conducted by combination of both quantitative analysis of the network structure and in-depth observation of the network formation processes, as described in the manuscript.

Modification in the supplementary information.

Page 2:

Network pattern classification. According to our quantitative image analyses, the network patterns can be classified based mainly on Pearson's correlation coefficient values, the standard deviation value and the island/void sizes of the agarose network. We can set the criteria of PCC value at 0.2 to differentiate the network patterns: If the value of Pearson's correlation coefficient is lower than 0.2, the composite hydrogels were classified into orthogonal or interactive type II; otherwise, interactive type I or III. The network patterns are further divided by use of the standard deviation value and the island/void sizes of the agarose network. Compared to the single-component agarose gel, the network can be classified into the interactive type I or II if those values are significantly smaller; otherwise, orthogonal or interactive type III (in the interactive type I, the agarose network obviously changes into the fibrous structure). It has not been confirmed whether this classification can be applicable as a universal guide because of some exceptional composite hydrogels and the small sample size. For example, the **BPmoc-F₃**/agarose hydrogel prepared by the pH decreasing protocol shows an intermediate network that exhibits characteristics of both orthogonal and interactive III networks. Therefore, the network classification should be carefully conducted by combination of both quantitative analysis of the network structure and in-depth observation of the network formation process, as described in the main text.

Comment 17:

Supplementary Figure 14 shows DBS-COOH/Alx488-Agarose as Orthogonal, but the body of the manuscript labels it as Interactive Type III.

Reply 17:

We corrected this mistake.

Reviewers' Comments:

Reviewer #1:

Remarks to the Author:

I am very happy with the revisions the authors have made to the manuscript in response to my suggestions (Reviewer 1). They have performed new experiments and gained new insights, and I am really pleased to see that some of the suggestions I made turned out to provide greater insight into these fascinating systems.

I have not considered the comments made by other reviewers, or the authors' responses to them. That is for the other reviewers to address.

Reviewer #2:

Remarks to the Author:

The authors have performed an array of additional experiments in revising their manuscript, including detailed rheological characterizations, pH-dependent assembly studies, and additional line-plots with statistical analysis. Each of these experiments improves the quality of the manuscript and answers several of the questions asked by the reviewers. The added explanation of the material design and discussion of the molecular structures of the peptide-type gelators also provides much needed clarity and background to this work. While the rheological characterization clearly revealed that GalNAc-cycC6/Agarose (interactive type 1) hydrogel possessed the highest storage and loss moduli, it is unclear why this would be the case, and the authors need to add some explanation here. Overall, the work is much improved and would be suitable for publication in Nature Communications after this last point is sufficiently addressed.

Reviewer #3:

Remarks to the Author:

The authors addressed most concerns of this reviewer. Therefore, I recommend acceptance of this revised manuscript.

Reviewers comments:

Reviewer 2 (Remarks to the Author):

Comment 1:

The authors have performed an array of additional experiments in revising their manuscript, including detailed rheological characterizations, pH-dependent assembly studies, and additional line-plots with statistical analysis. Each of these experiments improves the quality of the manuscript and answers several of the questions asked by the reviewers. The added explanation of the material design and discussion of the molecular structures of the peptide-type gelators also provides much needed clarity and background to this work. While the rheological characterization clearly revealed that GalNAc-cycC6/Agarose (interactive type 1) hydrogel possessed the highest storage and loss moduli, it is unclear why this would be the case, and the authors need to add some explanation here. Overall, the work is much improved and would be suitable for publication in Nature Communications after this last point is sufficiently addressed.

Reply 1:

We assume that the increase in the storage moduli and the brittleness may be caused by the high cross-link density and the thick and less flexible bundle formation between supramolecular and polymer fibrous network, as visualized by the CLSM imaging where each network is well overlapped to form a thick fibrous structure.

We modified the sentence to give some explanation for the highest storage and loss moduli of GalNAc-cycC6/Agarose (interactive type I).

Modification in the main text.

Page 11, line 4:

Taken together, the GalNAc-cycC₆/Agarose gel is stiffest and most brittle among the composite hydrogels we tested, presumably due to formation of denser crosslinks and/or thicker and less flexible bundles between two different networks. Our results suggest the network patterns of the composite hydrogels might give impacts on their rheological properties.